# Functional role of TRIM E3 ligase oligomerization and regulation of catalytic activity

Marios G Koliopoulos[1,†], Diego Esposito[1,†], Evangelos Christodoulou[2], Ian A Taylor[3] & Katrin Rittinger[1,*]

## Abstract

TRIM E3 ubiquitin ligases regulate a wide variety of cellular processes and are particularly important during innate immune signalling events. They are characterized by a conserved tripartite motif in their N-terminal portion which comprises a canonical RING domain, one or two B-box domains and a coiled-coil region that mediates ligase dimerization. Self-association via the coiled-coil has been suggested to be crucial for catalytic activity of TRIMs; however, the precise molecular mechanism underlying this observation remains elusive. Here, we provide a detailed characterization of the TRIM ligases TRIM25 and TRIM32 and show how their oligomeric state is linked to catalytic activity. The crystal structure of a complex between the TRIM25 RING domain and an ubiquitin-loaded E2 identifies the structural and mechanistic features that promote a closed E2~Ub conformation to activate the thioester for ubiquitin transfer allowing us to propose a model for the regulation of activity in the full-length protein. Our data reveal an unexpected diversity in the self-association mechanism of TRIMs that might be crucial for their biological function.

**Keywords** enzyme mechanism; protein structure; TRIM25; TRIM32; ubiquitin ligase

**Subject Categories** Post-translational Modifications, Proteolysis & Proteomics; Structural Biology

The EMBO Journal (2016) 35: 1204–1218

## Introduction

Tripartite motif (TRIM) E3 ligases constitute one of the largest subfamilies of RING E3s and regulate many cellular processes with a large proportion of TRIM family members being important in the regulation of innate immune responses (Meroni & Diez-Roux, 2005; Hatakeyama, 2011; Napolitano & Meroni, 2012; Rajsbaum et al, 2014). TRIM ligases share a conserved domain architecture that consists of an N-terminally located TRIM motif and a C-terminal region of variable composition that often contains protein interaction domains and may act as a substrate-recognition module. The tripartite motif consists of a RING domain, followed by one or two B-box domains and a coiled-coil (CC) region (Fig 1A) (Meroni & Diez-Roux, 2005). The RING domain of TRIMs adopts the canonical RING domain fold that recognizes the ubiquitin-loaded E2 conjugating enzyme (E2~Ub). B-boxes are zinc-binding motifs with a RING-like fold. They are divided into two types, termed B-box1 and B-box2, that coordinate the two zinc ions in two slightly different ways (Massiah et al, 2006; Tao et al, 2008). All TRIM ligases contain at least B-box2 while those with tandem B-boxes contain one of each with B-box1 located N-terminal to B-box2. B-box domains act as protein–protein interaction motifs and mediate self-association in some TRIMs, which may be important for higher order TRIM oligomerization as seen in the case of TRIM5α (Tao et al, 2008; Diaz-Griffero et al, 2009; Li et al, 2011).

E3 ubiquitin ligases are classed into three sub-families according to their catalytic mechanism: HECT and RBR-type ligases form a thioester intermediate with ubiquitin before its final transfer onto the substrate, though each sub-family uses different structural features to perform this function (Huibregtse et al, 1995; Wenzel et al, 2011; Smit & Sixma, 2014; Spratt et al, 2014). In contrast, RING E3 ligases act as adaptors to bring the ubiquitin-loaded E2 together with the substrate and promote ubiquitin transfer without directly participating in the reaction (Berndsen & Wolberger, 2014; Metzger et al, 2014). E2~ubiquitin intermediates exist in multiple conformations in the apo form that cycle between "open" and "closed" states. Some E2 conjugating enzymes, such as UBE2D (UbcH5) isoforms, adopt mostly open states whereas others, such as UBE2N (Ubc13), are more often found in the closed conformation (Pruneda et al, 2011, 2012). Recent work from a number of groups has shown that RING ligases induce a closed conformation of the otherwise dynamic E2~Ub intermediate and thereby prime the thioester bond for ubiquitin transfer (Dou et al, 2012, 2013; Plechanovova et al, 2012; Pruneda et al, 2012). Dimeric RINGs stabilize the closed E2~Ub conformation by using both RINGs in the dimer to contact the ubiquitin of the E2-intermediate simultaneously (Dou et al, 2012; Plechanovova et al, 2012). Monomeric RINGs can also stabilize a closed conformation; however, they do so using different structural elements outside the core RING domain (Dou et al, 2013).

1   Mill Hill Laboratory, Molecular Structure of Cell Signalling Laboratory, The Francis Crick Institute, London, UK
2   Structural Biology Science Technology Platform, The Francis Crick Institute, London, UK
3   Mill Hill Laboratory, Macromolecular Structure Laboratory, The Francis Crick Institute, London, UK
    *Corresponding author. Tel: +44 208 8162395; E-mail: katrin.rittinger@crick.ac.uk
    †These authors contributed equally to this work

    

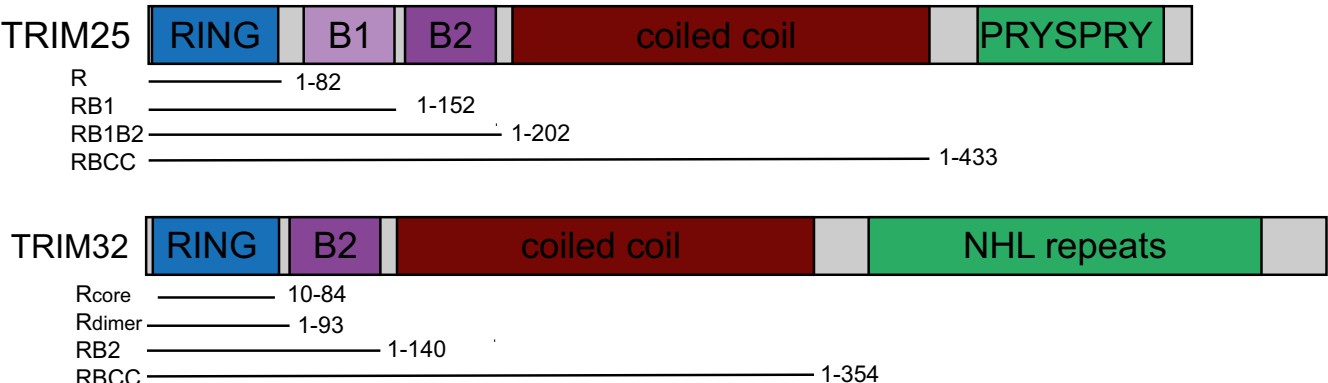

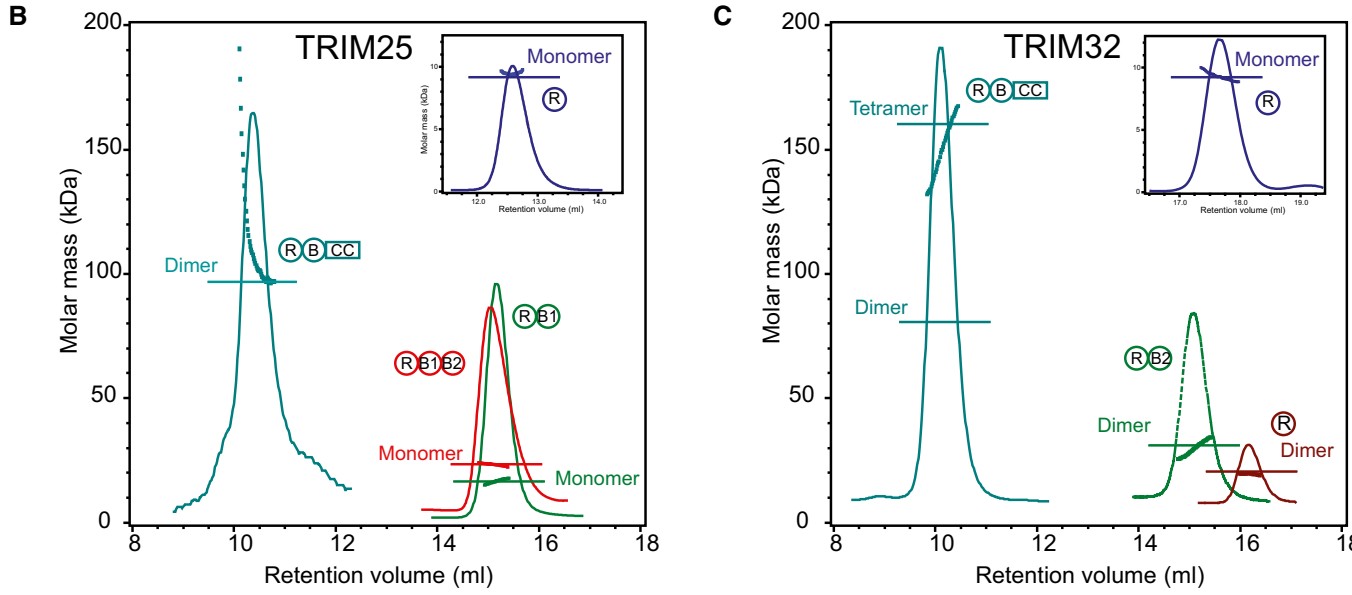

**Figure 1. Domain structures of TRIM25 and TRIM32 and oligomerization.**

A    Domain organization of TRIM25 and TRIM32 including fragments used in this study.

B, C  (B) SEC-MALLS traces of different TRIM25 and (C) TRIM32 constructs. The proteins are colour coded, and the domain architecture is reported next to the respective MALLS curve. The constructs were analysed over different concentration ranges, and the data are reported for: T25R, 4 mg/ml; T25RB1, 3.2 mg/ml; T25RB1B2, 3.6 mg/ml; T25RBCC, 0.5 mg/ml; T32R (core and extended), 3 mg/ml; T32RB, 4 mg/ml; T32RBCC, 4 mg/ml.

TRIM ligases dimerize through their coiled-coil region, and self-association has been proposed to be a requirement for catalytic activity (Streich *et al*, 2013), though a systematic analysis of the relationship between oligomerization and catalytic activity is currently missing. In addition, there have been suggestions that substrate binding might induce higher order oligomers that may further enhance activity (Yudina *et al*, 2015). Though structural details of the mechanistic features that could explain oligomerization-dependent activity are currently sparse, coiled-coil-mediated RING dimerization and subsequent stabilization of a closed E2~ubiquitin intermediate could explain this observation. However, recent structural studies showed that some TRIMs, and possibly all,

adopt an antiparallel coiled-coil orientation (Goldstone *et al*, 2014; Li *et al*, 2014; Sanchez *et al*, 2014; Weinert *et al*, 2015). Such an anti-parallel arrangement locates the N-terminal RING domains at opposite ends of the molecule implying that a dimeric RING arrangement may require formation of higher order TRIM complexes. For some TRIMs such as the retroviral restriction factor TRIM5α, higher order complexes have been shown to exist and constitute the active form (Diaz-Griffero *et al*, 2009; Li *et al*, 2011), whereas others, such as TRIM25, have been shown to be solely dimeric (Li *et al*, 2014).

To understand the role of self-association and gain insight into the mechanism of TRIM-mediated ubiquitination, we have analysed

the properties of two TRIM ligases, TRIM25 and TRIM32, which differ in their domain architecture with TRIM25 containing two B-boxes compared to a single B-box2 in TRIM32. TRIM25 plays a crucial role in the anti-viral response by ubiquitinating the N-terminal caspase activation and recruitment domains (CARDs) of the cytosolic pattern recognition receptor RIG-I. This event is critical to promote interaction with the adaptor protein MAVS and the subsequent activation of a signalling pathway leading to type 1 interferon production (Gack *et al*, 2007). TRIM32 has a broader cellular role and has been shown to regulate neuronal development, the stability of sarcomeric architecture and has more recently been reported to restrict influenza A virus (Shieh *et al*, 2011; Zhang *et al*, 2012; Fu *et al*, 2015). Both TRIM ligases work with UBE2N/UBE2V1 (Ubc13/Uev1) to form K63-linked chains which play a non-degradative role, but are also active with UBE2D isoforms (Zeng *et al*, 2010; Napolitano *et al*, 2011; Zhang *et al*, 2012).

We have used a combination of structural, biophysical and biochemical approaches to describe the oligomeric state of TRIM25 and TRIM32 and determine the role of individual domains in self-association. We show that their oligomerization mode differs significantly and present a model for its link with catalytic activity and their physiological function. Furthermore, we provide mechanistic insight into the ubiquitination reaction and identify structural features that are important for ubiquitin transfer.

# Results

## Role of tripartite motif domains in mediating self-association

To assess how individual domains within the tripartite motif of a given TRIM regulate its oligomeric state, we analysed multiple TRIM25 (Fig 1A and B) and TRIM32 (Fig 1A and C) constructs by analytical size-exclusion chromatography coupled with multi-angle laser light scattering (SEC-MALLS). Samples were run over a range of concentrations and the oligomeric state of the proteins determined (Fig 1B and C). We obtained an essentially symmetric chromatographic light-scattering profile consistent with a single molar mass for all TRIM25 and TRIM32 constructs except for TRIM25 RBCC, which has a propensity to form higher molecular weight species with increased concentration. In agreement with its NMR solution structure (2CT2.pdb, Riken Structural Genomics/Proteomics Initiative), the core TRIM32 RING construct elutes as a monomeric species (Fig 1C). In contrast, a fragment that is extended by 10 and 9 residues N- and C-terminally, which are predicted to be α-helical, is dimeric at all concentrations tested, suggesting that TRIM32 might dimerize in a manner similar to that of a number of homo- and heterodimeric RING ligases such as the BRCA1/BARD1 complex, where N- and C-terminally located α-helices form a four-helix bundle (Brzovic *et al*, 2001). The presence of the B-box domain does not change the oligomeric state of TRIM32. In contrast to the dimeric nature of the extended TRIM32 RING construct, all fragments of TRIM25 that contain the RING and B-box domains are monomeric in the range of concentrations explored (Fig 1B and C).

The crystal structures of the coiled-coil regions of TRIM5α, TRIM20, TRIM25 and TRIM69 all show an anti-parallel dimer for this region, and based on sequence conservation, this behaviour might be a general property of the TRIM family (Goldstone *et al*,

2014; Li *et al*, 2014; Sanchez *et al*, 2014; Weinert *et al*, 2015). In support of this suggestion, inclusion of the CC region mediates further self-association and we detect a single tetrameric species with a molecular weight of ~160 kDa for a fragment of TRIM32 (RBCC) in which the dimeric RING/B-box construct is extended to include the coiled-coil (Fig 1C). As described above, extension of the TRIM25 RB1B2 fragment to include the CC region makes the protein prone to oligomerization and the molecular mass detected for the RBCC construct is consistent with the formation of dimers and higher molecular weight species at increasing concentrations (Fig 1B). However, higher order association is likely to be an artefact of the missing C-terminus as the isolated coiled-coil region and the full-length protein that includes the C-terminal PRYSPRY (also called B30.2) domain have both been shown to be dimeric (Li *et al*, 2014; Sanchez *et al*, 2014).

## The role of tripartite motif domains and oligomerization in catalytic activity

To understand how the observed oligomeric state of TRIM ligases relates to catalytic activity, we carried out ubiquitination assays with different TRIM25 and TRIM32 constructs in conjunction with UBE2D3 isoforms that mediate TRIM auto-ubiquitination in the absence of another substrate and with heterodimeric UBE2N/UBE2V1 that catalyses formation of unanchored K63-linked poly-ubiquitin chains. Auto-ubiquitination assays showed large apparent differences in the activity of RING-only (R), RING/B-box (RB1 and RB1B2) and RING/B-box/CC (RBCC) constructs of both TRIM25 and TRIM32 (M.G. Koliopoulos & D. Esposito, unpublished observation). However, because the constructs tested are of different length, they contain varying numbers of lysine residues available for modification making it difficult to directly compare their relative activity using this assay. Therefore, we used lysine-discharge assays, which monitor the ability of a given construct to activate a pre-charged E2~Ub intermediate for transfer of ubiquitin onto free lysine. In these assays, the monomeric R, RB1 and RB1B2 fragments of TRIM25 were all able to enhance discharge of the UBE2D~Ub intermediate, with a slight increase in activity of the longer fragments (Figs 2A and B, and EV1). By comparison, the reaction is significantly enhanced in the presence of the entire RBCC motif (Figs 2A and B, and EV1). These data would suggest that the RING of TRIM25 is active as a monomer but that CC-induced dimerization increases activity. In contrast, we could not detect activity for the monomeric TRIM32 core RING domain. However, the extended dimeric RING showed a similar level of activity as the B-box-containing fragment, which was further enhanced in the tetrameric TRIM32 RBCC construct albeit to a much lower extend compared to TRIM25.

The UBE2N/UBE2V1 complex synthesizes the formation of unanchored K63-linked Ub chains even in the absence of an E3 ligase, though the presence of an E3 significantly enhances the reaction. This heterodimeric E2 has been reported to be one of the cognate E2s for TRIM25 and TRIM32, and thus, we used it to compare their ability to catalyse the formation of poly-ubiquitin chains independent of the nature of the substrate (Zeng *et al*, 2010; Napolitano *et al*, 2011; Zhang *et al*, 2012). As observed in the discharge assays, the monomeric TRIM25 RING-only and B-box-containing fragments are active (Figs 2E and F, and EV1), whereas TRIM32 only showed

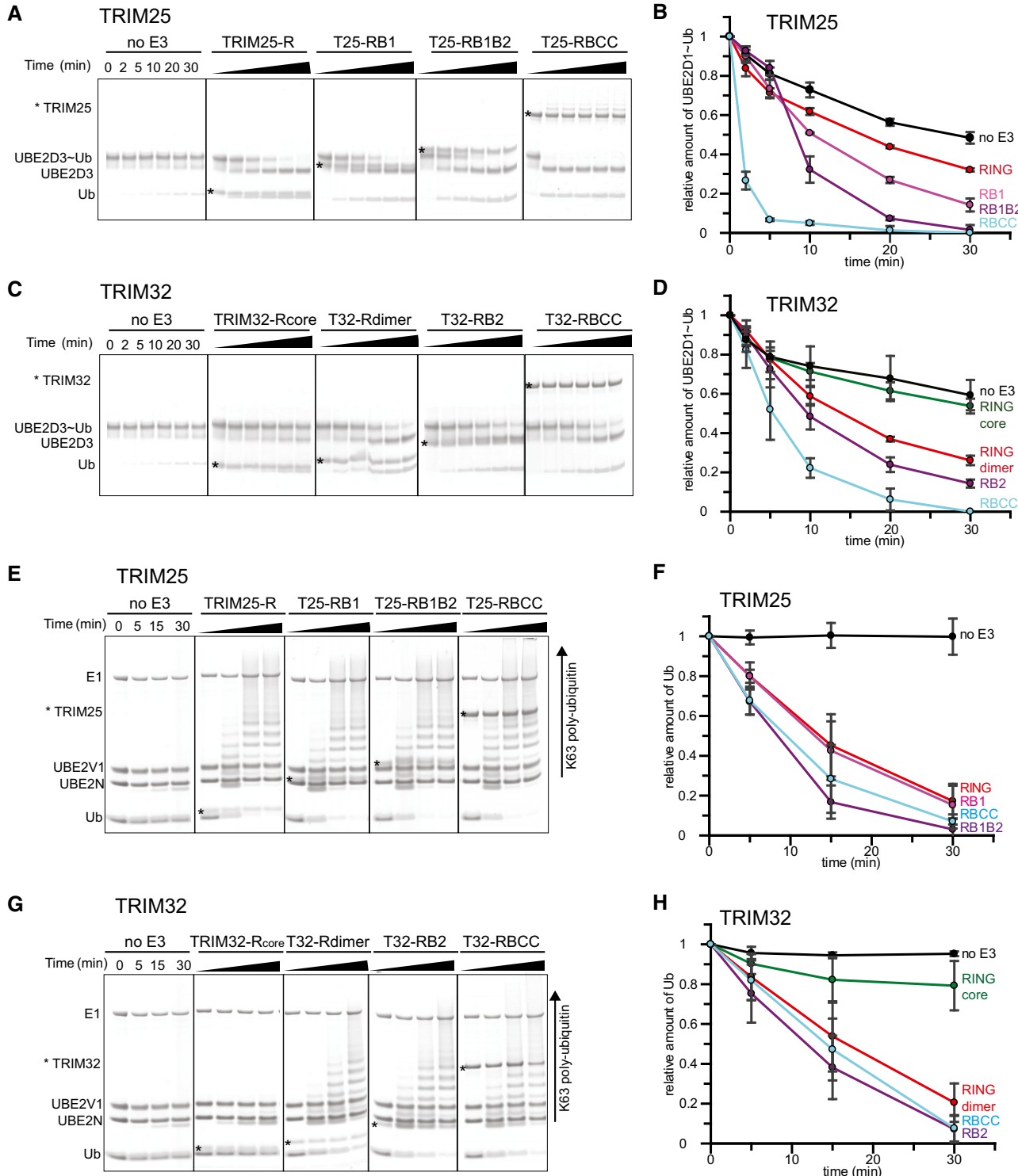

**Figure 2.  Relationship between oligomeric state and catalytic activity.**

A–D  (A, C) UBE2D3~Ub discharge assays with different TRIM25 and TRIM32 constructs, respectively. Assays were carried out with TRIM constructs as indicated and the reaction was monitored over 30 min. The asterisk indicates the band for the TRIM construct. (B, D) Quantification of the discharge assays using UBE2D1~Ub$^{Atto}$ and TRIM25 or TRIM32, respectively. The loss of UBE2D1~Ub$^{Atto}$ is plotted as the average of experimental triplicates (± s.d.).

E–H  (E, G) K63 poly-ubiquitination assays using UBE2N/UBE2V1 and TRIM25 and TRIM32, respectively. Reactions were incubated for 30 min and samples taken at the indicated times. The asterisk indicates the band for the TRIM construct. (F, H) Assays were quantified (see Materials and Methods) by supplementing the reaction with 1 μM Ub$^{Atto}$ and integrating the loss of free Ub$^{Atto}$ for TRIM25 and TRIM32, respectively. Assays were carried out in triplicate (± s.d.).

Source data are available online for this figure.

activity for dimeric constructs (Figs 2G and H, and EV1). Interestingly, there are no significant differences in the activity of different constructs tested in conjunction with UBE2N/UBE2V1, in contrast to the discharge assays in which especially the TRIM25 RBCC construct showed a significantly higher level of activity than the shorter fragments.

Taken together, these data confirm that the RING is the main catalytic unit of TRIMs and indicate that B-boxes only make minor contributions to catalytic activity. Most surprisingly, these results suggest that the RING of TRIM32 is only active as a dimer whereas the RING of TRIM25 does appear not to require dimerization for catalytic activity, in contrast to previous suggestions that TRIMs require self-association for activity. To gain molecular insight into these contrasting mechanisms of ubiquitin transfer, we determined the crystal structures of the RING domains of TRIM25 and TRIM32 with and without an ubiquitin-loaded E2, respectively.

### The crystal structure of the dimeric TRIM32 RING domain

The extended RING domain of TRIM32 crystallized as a dimer with a significant proportion of the dimer interface formed by two α-helices located N- and C-terminal to the core RING domain, respectively, that form a four-helix bundle (Fig 3A). Dimerization buries 970 $A^2$ and the dimer interface is formed predominantly by hydrophobic residues located within the four helices. RING dimerization through a four-helix bundle employing regions outside the RING has been observed in a number of dimeric RINGs such as the BRCA1/BARD1 heterodimer or the Rad18 homodimer (Brzovic *et al*, 2001; Huang *et al*, 2011). Furthermore, this arrangement is similar to that recently reported in the crystal structures of the RING domains of the retroviral restriction factor TRIM5α (Yudina *et al*, 2015) (Fig 3B) and TRIM37 (Northeast Structural Genomics Consortium, 3LRQ.pdb).

### The crystal structure of the TRIM25 RING/UBE2D1~Ub complex

The TRIM25 RING/UBE2D1~Ub complex crystallized with two molecules of the TRIM25 RING domain and two molecules of a stable, isopeptide-mimic of the UBE2D1~Ub intermediate in the asymmetric unit (AU), related by a twofold symmetry axis (Fig 3C) (Table 1). Rather surprisingly and in contradiction to our SEC-MALLS, the RING domain is a dimer in the crystal with the dimer

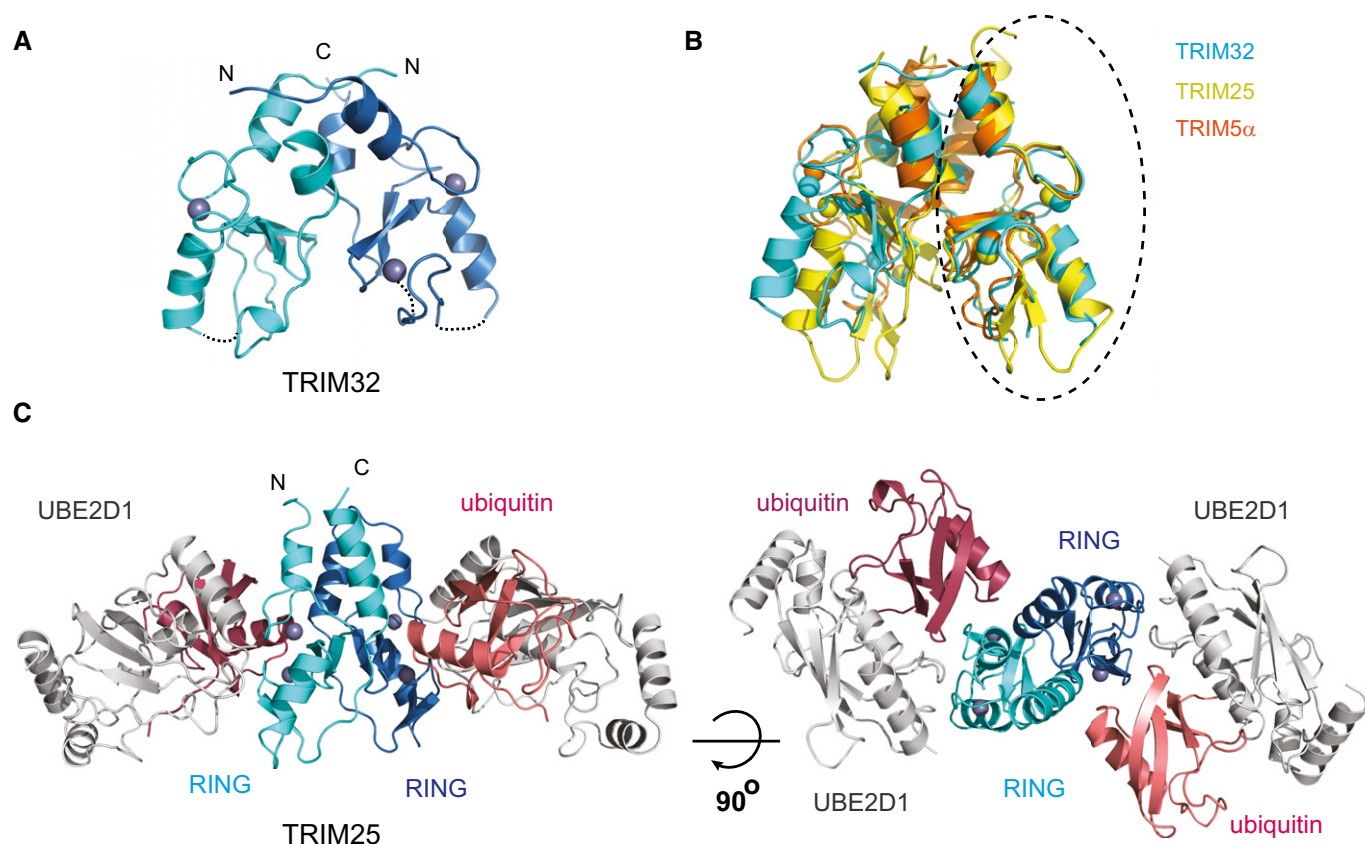

**Figure 3.  Structure of the RING dimers and interaction with the E2~Ub intermediate.**

A   Structure of the TRIM32 RING dimer in ribbon representation with each RING monomer coloured in cyan and blue and the Zn$^{2+}$ ions as grey spheres.
B   Overlap of the RING dimers of TRIM32 (cyan, 5FEY.pdb), TRIM25 (yellow, 5FER.pdb) and TRIM5α (orange, 4TKP.pdb). The structures were overlapped on the circled RING domain. This overlap shows that the structures of the RINGs are very similar but that there are differences in the relative orientations of the two RINGs.
C   Structure of the TRIM25 RING/E2~Ub complex with the RING domains in the same colour scheme as in (A), UBE2D1 in grey and the ubiquitin molecules in salmon and red.

interface burying ~1,000 Å$^2$ of solvent-accessible surface. As observed in TRIM32, dimerization is mediated by two α-helices flanking the RING domain. Structural comparison of the dimeric RINGs of TRIM25, TRIM32 and TRIM5α shows that individual monomers overlap well but that the precise arrangement of the helices and the relative angle of the RING domains in the dimer differ in different TRIMs (Fig 3B). Similar to TRIM32, the dimer interface is formed predominantly by hydrophobic residues (L4, L7, L11, V68, L69, V72 and F76) that direct their side chains into the interior of the four-helix bundle (Fig 4A). Contacts between the core RING domains are mediated by residues preceding the adjacent C-terminal α-helix and of the β-sheet. The two RING/E2~Ub complexes in the AU are highly similar and overlap with an rms deviation of 0.37 Å with the main difference being the termini of the helices adjacent to the RING which contain a few more ordered residues in one copy.

To gain insight into a potential role of dimerization in enzymatic activity of TRIM25, we mutated V72, which is located at the centre of the four-helix bundle to arginine (Fig 4A). Mutation of the equivalent residue has previously been shown to interfere with catalytic activity of TRIM5α, another TRIM ligase whose RING is monomeric in solution but also crystallizes as a dimer (Yudina *et al*, 2015).

**Table 1.  Data collection and refinement statistics.**

| Crystal | TRIM32 RING | TRIM25/UBE2D1~Ub |
|---|---|---|
| Resolution | 38.56–2.23 (2.31–2.23) | 30.28–2.34 (2.42–2.34) |
| Space group | P4$_3$2$_1$2 | P2$_1$2$_1$2$_1$ |
| Cell dimensions | | |
| a,b,c (Å) | 54.54, 54.54, 99.12 | 60.48, 71.70, 160.43 |
| α, β, γ (°) | 90.0, 90.0, 90.0 | 90.0, 90.0, 90.0 |
| $R_{merge}$ | 0.1401 (0.962) | 0.139 (1.918) |
| Total reflections | 161,599 | 384,013 |
| Unique reflections | 7,770 | 30,000 |
| Redundancy | 20.8 | 12.8 |
| Completeness (%) | 99.63 | 99.32 |
| <I/σ(I)> | 13.28 (1.81) | 13.75 (1.53) |
| CC$_{1/2}$ | 0.996 (0.782) | 0.998 (0.523) |
| Refinement | | |
| $R_{work}$ | 0.248 | 0.227 |
| $R_{free}$ | 0.273 | 0.273 |
| Number of atoms | 1,254 | 4,832 |
| Protein | 1,223 | 4,760 |
| Zn$^{2+}$ | 4 | 4 |
| Water | 27 | 68 |
| Average B-factor (Å$^2$) | 85.6 | 56.0 |
| RMS bonds (Å) | 0.002 | 0.002 |
| RMS angles (°) | 0.486 | 0.539 |
| Ramachandran statistics | | |
| Favoured region (%) | 93.2 | 96.11 |
| Allowed region (%) | 6.8 | 3.38 |
| Disallowed region (%) | 0.0 | 0.51 |

Data in the highest resolution shell are shown in parentheses.

Given that we were not able to detect dimerization in solution and hence were not able to test whether the V72R indeed disrupted the ability of TRIM25 RING to dimerize, we also produced a tandem RING–RING construct, in which the two RING domains are covalently connected by a short linker, which would be expected to stabilize the weak interface and form a constitutive dimer ("RING-linker-RING"). The V72R mutant protein behaved like the monomeric core RING construct of TRIM32 and showed no significant activity in discharge assays and in ubiquitination assays with UBE2N/UBE2V1 (Figs 4B–D and EV2). In contrast, the covalently linked RING dimer was able to discharge the E2~Ub adduct much faster compared to the monomeric RING domain and all E2~Ub thioester was discharged already after 2 min (Fig 4B and C). Introduction of the V72R mutant into the covalently linked RING dimer significantly reduced catalytic activity but not to the same extent as in the monomeric RING. These data suggest that the activity observed in assays using the monomeric RING constructs must result from a small proportion of dimeric species that is present under assay conditions. Given that no dimer could be detected during SEC-MALLS experiments, this suggests that dimerization might be induced by binding of the E2~Ub intermediate. In contrast, synthesis of K63-linked chains by UBE2N/UBE2V1 was not enhanced by the fused RING dimer in agreement with our previous data that showed that oligomerization does not have a significant effect on the formation of free K63 chains (Fig 4D). At present, it is not clear what the molecular basis for this different behaviour is, but we speculate that it is linked to the observation that UBE2N~Ub intermediates populate the "closed" active state to a much higher degree in the absence of an E3 than do UBE2D~Ub intermediates (Pruneda *et al*, 2011).

**Mechanism of ubiquitin transfer**

Each RING monomer contacts a single UBE2D1 molecule via the conserved E2–E3 interface that has been observed in other E2–RING complexes. The structures of two dimeric RING E3s in complex with UBE2D-Ub have been reported recently, and both show that the proximal RING binds the ubiquitin intermediate via ubiquitin's I36 surface patch. An aromatic residue from the distal RING makes additional contacts with the same ubiquitin surface, thereby stabilizing a "closed" conformation in which the E2~Ub is activated for transfer (Dou *et al*, 2012; Plechanovova *et al*, 2012). Similarly, in the TRIM25/E2~Ub structure each ubiquitin molecule is folded back onto its E2 in a conformation that resembles the one seen in other dimeric RING/E2~Ub complexes. In this closed conformation, I44 of ubiquitin packs against L104 of UBE2D1. Similarly, R54 of TRIM25, referred to as the linchpin for allosteric activation of E2~Ub (Pruneda *et al*, 2012; Metzger *et al*, 2014), is involved in electrostatic interactions with Q92 of UBE2D1 and Q40 and R72 of ubiquitin, and accordingly, its mutation reduces catalytic activity in discharge assays and formation of K63-linked chains by UBE2N/UBE2V1 (Fig 4B–D). In contrast, the aromatic residue from the distal RING (Y193 of RNF4 and F296 in BIRC7), that is important in other RINGs to engage ubiquitin in the closed conformation, does not exist in TRIM25 (Dou *et al*, 2012; Plechanovova *et al*, 2012). Instead, residues K65 and T67 of the RING domain contact D32 and E34 of the opposite, distal ubiquitin in one monomer, whereas in the other monomer, the K65 and D32 interaction is absent and

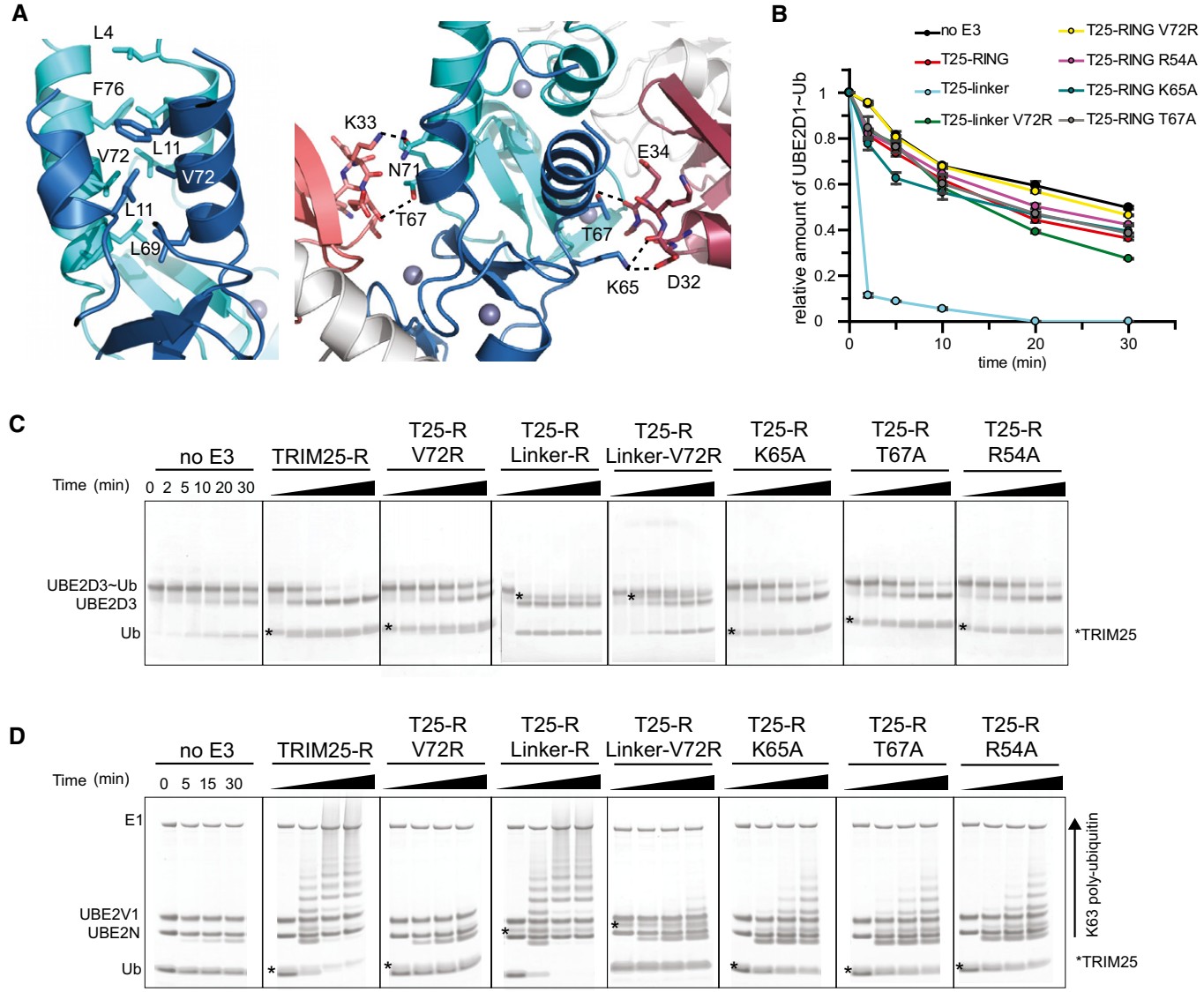

**Figure 4.  RING dimerization and the interaction with ubiquitin.**

A   Close-up of the TRIM25 RING dimer interface highlighting the hydrophobic interactions made between the four α-helices (left). Close-up of the interface between each RING monomer and the proximal ubiquitin (right).

B   UBE2D1~Ub$^{Atto}$ discharge assays with TRIM25 wild-type RING, the fused RING constructs and different mutants important for dimerization or the interaction with ubiquitin. Time point zero for the T25-R Linker and T25-R Linker V72R samples was taken before the addition of E3 as discharge is very fast. The loss of UBE2D1~Ub$^{Atto}$ is plotted as the average of experimental duplicates ($\pm$ s.d.).

C   UBE2D3~Ub discharge assays with the same mutants as in (B), stained with InstantBlue.

D   K63 poly-ubiquitination assays using UBE2N/UBE2V1. The asterisk indicates the band for the TRIM construct.

Source data are available online for this figure.

replaced by a hydrogen bond between N71 of the RING and K33 of ubiquitin (Fig 4A). Mutation of either K65 or T67 to alanine reduces catalytic activity (Figs 4B–D and EV2).

This structure establishes that TRIM25 uses a dimeric RING/E2~Ub arrangement, in which both RINGs contact ubiquitin, to activate the ubiquitin thioester intermediate for ubiquitin transfer. However, close inspection of the structure shows that TRIM25 uses an additional feature to maintain the closed conformation, that is reminiscent of the mechanism identified in the monomeric E3 ligase

Cbl-b, where a phospho-tyrosine (pTyr) located outside the RING domain makes additional contacts with ubiquitin to stabilize the closed conformation (Dou *et al,* 2013). A similar arrangement is present in TRIM25; however, in this case a charged residue, E10, located in the N-terminal helix forms a network of electrostatic interactions and hydrogen bonds that stabilizes the observed RING/E2~Ub arrangement. In one monomer, E10 of the RING contacts K11 of the proximal ubiquitin while simultaneously engaging N71 of the opposite RING, which in turn forms a hydrogen bond with the

same ubiquitin (Fig 5A). In the other monomer, the hydrogen bond between E10 from one RING to N71 of the other is present, but the interactions with K11 and K33 are absent. To investigate the importance of E10 in stabilizing the observed conformation, we tested the catalytic activity of the E10R mutant. Due to its close proximity to E10, we also examined the role of E9 in catalysis. Mutation of E10R completely abolished the catalytic ability of T25R to discharge UBE2D~Ub and to form K63-linked ubiquitin chains with UBE2N/UBE2V1 (Fig 5B and D). In contrast, E9R discharges UBE2D~Ub

even more efficiently than wild-type RING (Fig 5B and D). These data clearly establish that the dual function of E10, to contact the opposite RING and ubiquitin, is necessary for overall activity. This observation gains further significance when taking sequence alignments into account that indicate that the acidic motif with either one or two Glu residues is conserved in many TRIM ligases (Li *et al*, 2014). In support of a more general role of this acidic residue, mutation of the equivalent E16 in TRIM32 also disrupts catalytic activity (Fig 5C and E). This effect is similar to the mutation of I85R, which

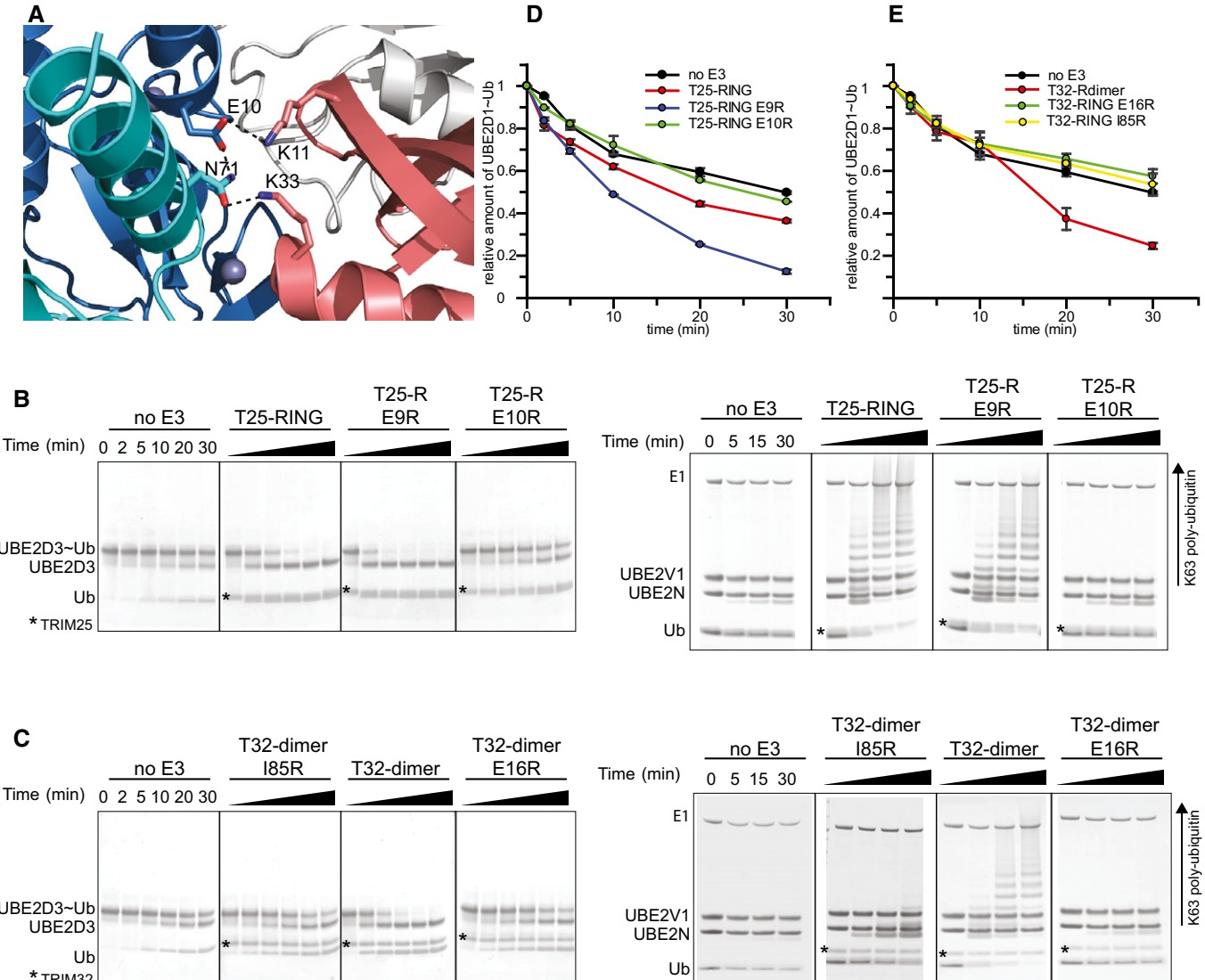

**Figure 5.  Stabilization of the closed E2~Ub conformation.**

A   Role of E10 in the RING of TRIM25 (blue) in stabilizing the closed E2~Ub conformation by contacting K11 from the proximal ubiquitin (salmon) and N71 of the opposite RING (cyan) which in turn contacts K33 of the same ubiquitin.

B   E2~Ub discharge and K63 poly-ubiquitination assays to test the role of E9 and E10 in TRIM25 activity. Substitution of Glu9 with Arg has no significant effect on activity, whereas the E10R mutation almost completely abolishes catalytic activity.

C   Discharge and K63 poly-ubiquitination assays to test the role of the equivalent residue E16 in TRIM32 and the role of I85R. Mutation of E16R abolishes catalytic activity indicating that the role of the glutamate is conserved.

D, E   Quantification of UBE2D1~Ub$^{Atto}$ discharge assays. The loss of E2~Ub is plotted as the average of experimental duplicates (± s.d.).

Source data are available online for this figure.

interferes with dimerization, similar to V72R in TRIM25 (Figs 5C and E, and EV2). Moreover, structural and sequence comparison of the RINGs of TRIM25 and TRIM5α shows that E11 and E12 occupy similar positions (Yudina *et al*, 2015). Intriguingly, structural alignment of the RINGs of TRIM25 and monomeric Cbl-b shows that Y363 of Cbl-b, which needs to become phosphorylated for full ligase activity, is in the equivalent position of E10, although the residues come from different structural elements (Dou *et al*, 2013). This suggests that both monomeric and dimeric E3s may make use of an acidic element to stabilize the active E2~Ub conformation.

### Self-association of the TRIM25 RING domain in solution

Given the apparent difference in self-association of the TRIM25 RING domain in solution versus our crystal structure, we further explored its ability to dimerize. To do so, we compared the NMR characteristics of the RING domains of TRIM25 and TRIM32. The 2D $^1$H-$^{15}$N-HSQC spectra of the TRIM25 RING construct were recorded at concentrations of 125 μM (Fig 6A), 570 μM (Fig 6B) and 1 mM and compared to the spectra of the TRIM32 RING$_{core}$ monomer (Fig 6C) and TRIM32 RING dimer (Fig 6D). The cross-peaks in the 2D $^{15}$N-HSQC spectra of TRIM25 RING at different concentrations have varied intensities and linewidths compared to the well-dispersed and sharp peaks in the spectrum of the monomeric TRIM32 RING$_{core}$. No chemical shift perturbations of the well-dispersed resonances occur at increasing concentration of TRIM25 RING, indicating that the overall structure of the domain is retained and that the increase in linewidth observed is likely due to an exchange between a monomeric and a dimeric form of the protein.

We further characterized the hydrodynamic properties of the RINGs of TRIM25 and TRIM32 by $^{15}$N-nuclear relaxation rate measurements. The value for the isotropic rotational correlation time of the TRIM25 RING at lower concentration is $6.42 \pm 0.07$ ns (obtained from 35 well-resolved cross-peaks in the corresponding 2D $^1$H-$^{15}$N HSQC spectrum with a heteronuclear NOE value $I_{sat}/I_0 > 0.7$) (Fig 6A and E). This is similar to the value of 6.65 ns calculated from our X-ray crystal structure coordinates of one TRIM25 RING chain using HYDRONMR and to the value of 6.80 ns calculated from the coordinates for the monomeric TRIM32 RING$_{core}$ (2CT2.pdb). The value for the correlation time of the TRIM25 RING at higher concentration increases to $7.67 \pm 0.09$ ns at 570 μM and $9.3 \pm 0.1$ ns at 1 mM (Fig 6E), indicating an increase in molecular size due to dimerization. Notably, spin relaxation analysis of the TRIM32 RING domain results in a value significantly longer ($\tau_c = 15.2 \pm 0.2$ ns) than the one calculated from the dimeric crystal structure ($\tau_c = 11.3$ ns). This is most likely due to the tendency of the crystallized TRIM32 RING construct to self-associate at higher

concentration, which however is absent in longer constructs and likely an artefact of domain deletion. Taken together, our NMR data provide evidence for an inherent ability of the RING domain of TRIM25 to dimerize, which is low in the isolated RING domain but likely enhanced in the context of the full-length protein or by additional binding partners such as the E2~Ub intermediate.

### The structure of TRIM25 and TRIM32 in solution

To gain low-resolution structural information on all TRIM25 and TRIM32 constructs analysed in this study, small-angle X-ray scattering (SAXS) experiments were performed. Details of data collection and analysis are presented in Table 2. Guinier analysis of the SAXS profiles and *ab initio* modelling of the molecular envelopes for the various TRIM proteins reproduces the pattern of oligomerization observed by SEC-MALLS (Fig 1B and C). The core TRIM32 monomeric RING (T32Rcore) and RING of TRIM25 (T25R) show virtually identical scattering profiles and pair-distribution functions indicating a similar overall structure. The two proteins have slightly prolate envelopes with a maximum dimension of 48 Å and a cross section corresponding to the maximum of the P(r) distribution of ~16 Å (Fig 6F). The dimeric TRIM32 RING (T32R) has a slightly larger $D_{max}$ of 60 Å and the maximum of the P(r) function is shifted to the right, indicating a larger cross section as expected from a symmetric dimeric protein. The modelled molecular envelope has a larger lobe that accommodates the dimeric RING and a smaller lobe that likely accounts for the dimerization helices (Figs 6F and EV3). The molecular masses derived from the SAXS data are in good agreement with those calculated from the primary sequences (see Table 2). Fitting of the experimental scattering amplitudes to the available structural coordinates by the program CRYSOL (Svergun *et al*, 1995) shows a good agreement for the NMR structure of the TRIM32 RING monomer (PDB 2CT2: $\chi^2 = 0.4$) and the crystal structure of the TRIM32 RING dimer ($\chi^2 = 0.5$) (Fig EV3), implying that both sets of coordinates are a good representation of the monomeric and dimeric state of the protein in solution.

The P(r) distribution of TRIM25 containing both B-box domains clearly shows an elongated protein with a long $D_{max}$ value and a peak of the pair-distribution function at a distance value comparable to that of the monomeric RING, indicating an arrangement of adjacent domains similar to beads on a string (Fig 6G). Interestingly, the difference in $D_{max}$ between TRIM25 RING/B-box1 (RB1) and TRIM25 RING/B-box1+2 (T25RB1B2) is about 16 Å, which corresponds to the diameter of the globular RING domain. Given the short length of the linker between the two B-boxes, this observation points towards a restricted intra-domain mobility in an arrangement that is reminiscent of that of the tandem B-boxes of TRIM18, which pack against

---

**Figure 6.  Solution structure of TRIM25 and TRIM32.**

A–D    2D $^1$H-$^{15}$N-HSQC spectra of the TRIM25 RING domain at a concentration of 125 μM (A) and 570 μM (B), the monomeric TRIM32 RING$_{core}$ (C) and TRIM32 RING dimer (D).
E       Isotropic rotational correlation times of the different TRIM25 and TRIM32 RING constructs, obtained from the relaxation analysis of resonances in the corresponding 2D $^1$H-$^{15}$N HSQC spectra (dark grey) or calculated from available structures by HYDRONMR (light grey). Error bars are derived from relaxation rate analysis implemented in TENSOR2 (Dosset *et al*, 2000).
F       Scattering profiles of RING constructs (left) and their normalized pair-distribution functions P(r) (middle). The right-hand panel shows low-resolution *ab initio* models derived from the SAXS data analysis.
G       Scattering profiles, their normalized pair-distribution functions P(r) and low-resolution *ab initio* models for RING and B-box-containing constructs.
H       Scattering profiles, their normalized pair-distribution functions P(r) and low-resolution *ab initio* models for the RBCC domains of TRIM25 and TRIM32. The curves and envelopes are reported using the same colour scheme in all respective panels and the envelopes for the constructs are drawn to scale.

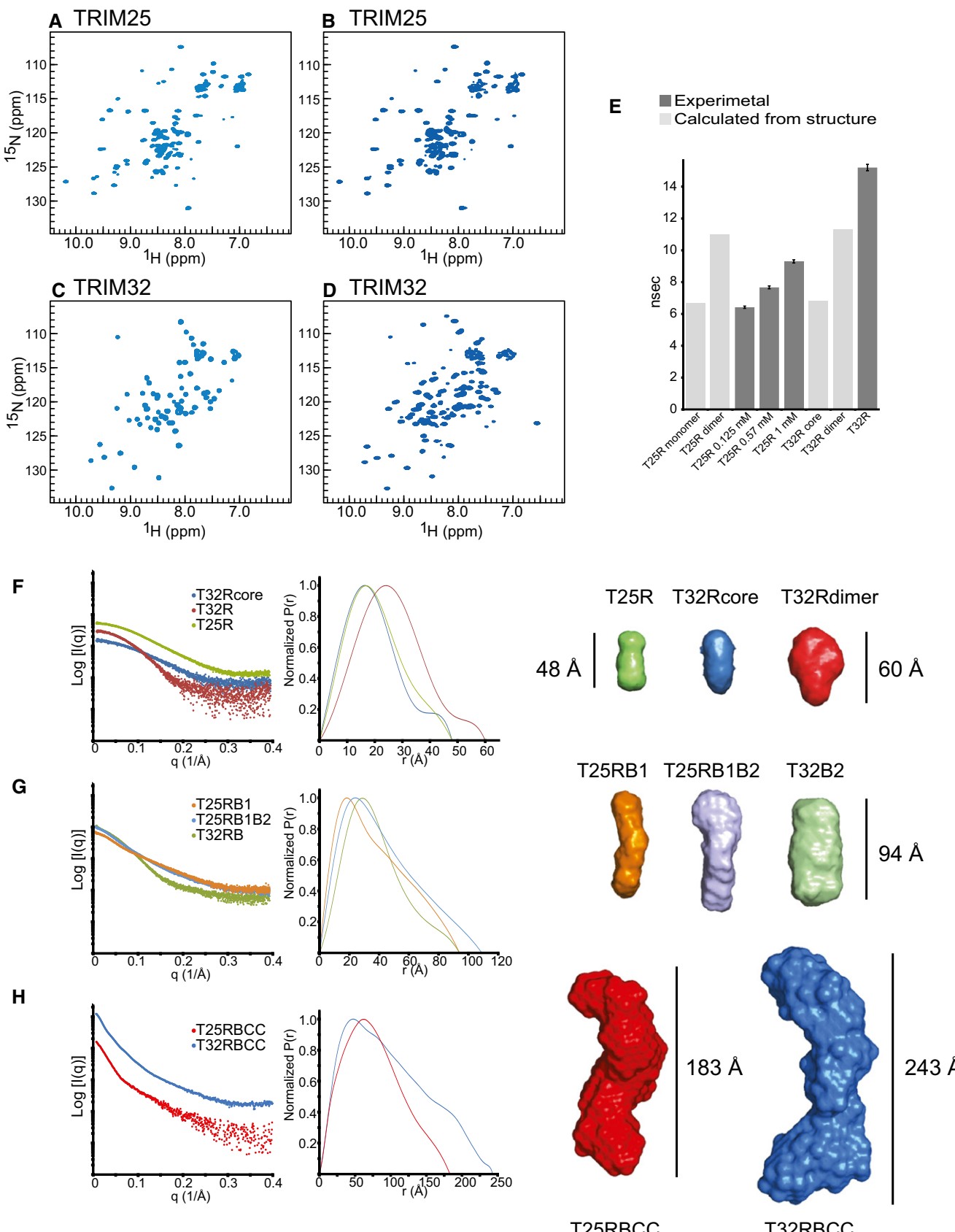

**Figure 6.**

**Table 2.  SAXS data analysis and modelling.**

| Construct | $R_g$ P(R)/Guinier (Å) | $D_{max}$ (Å) | MW (kDa) | MW from SAXS (kDa) | Porod volume (Å³) | $\chi^{2a}$ | NSD[b] |
|---|---|---|---|---|---|---|---|
| T32Rcore (2.0 mg/ml) | 15.38 ± 0.23/15.5 ± 0.52 | 48 | 9.4 | 6.8 | 18,800 | 0.28 | 0.53 ± 0.04 |
| T32R (2.0 mg/ml) | 20.04 ± 0.16/20.0 ± 0.24 | 60 | 11.0 | 19.6 | 59,000 | 0.32 | 0.58 ± 0.08 |
| T32RB2 (2.5 mg/ml) | 29.63 ± 0.093/27.4 ± 0.08 | 94 | 15.6 | 33.8 | 86,100 | 0.84 | 0.63 ± 0.12 |
| T32RBCC[c] (6 mg/ml) | 78.72 ± 0.07/78.29 ± 0.2 | 243 | 39.5 | 167 | 424,000 | 2.53 | 0.94 ± 0.13 |
| T25R (5.0 mg/ml) | 16.0 ± 0.04/15.8 ± 0.1 | 48 | 9.1 | 7.1 | 19,600 | 0.55 | 0.53 ± 0.03 |
| T25RB1 (3.5 mg/ml) | 31.1 ± 0.14/31.09 ± 0.08 | 94 | 16.7 | 11.3 | 42,500 | 0.82 | 0.61 ± 0.04 |
| T25RB1B2 (3.5 mg/ml) | 33.75 ± 0.092/33.73 ± 0.3 | 109 | 22.4 | 19.2 | 66,000 | 0.79 | 0.68 ± 0.04 |
| T25RBCC[c] (2.5 mg/ml) | 60.56 ± 0.085/60.3 ± 0.8 | 183 | 48.5 | 94.6 | 225,000 | 3.10 | 0.83 ± 0.10 |

[a]$\chi^2$ values derived from DAMMIF.
[b]Normalized Spatial Discrepancy.
[c]Data collected at Soleil, Gif-sur-Yvette. Other data were collected at Diamond Light Source, Oxford.

one another (Tao *et al*, 2008). In contrast, TRIM32 RING/B-box2 (T32RB2) has an envelope with a larger diameter and appears to be more symmetric as a result of its dimeric nature. The SAXS-derived molecular weight for T32RB2 (33.8 kDa) shows that the protein is dimeric in line with SEC-MALLS data (Fig 6G and Table 2).

Both TRIM constructs containing the RING, B-box and coiled-coil region are represented by very elongated proteins with pair-distribution functions that are reminiscent of cylindrical arrangements: a large maximum distance and a primary peak at much shorter *r* values (Fig 6H). These profiles are very similar to those reported for the coiled-coil region-containing fragments of TRIM5α and TRIM20 (Goldstone *et al*, 2014; Weinert *et al*, 2015). Remarkably, the maximum dimension of the envelope for the TRIM25 RBCC construct (183 Å) is not much longer than that for the isolated CC structure of TRIM25 (170 Å) and comparable to the lengths of the TRIM5α BCC (180 Å) and TRIM20 CC/B30.2 constructs (167 Å) (Fig EV3) (Goldstone *et al*, 2014; Sanchez *et al*, 2014; Weinert *et al*, 2015). Furthermore, dimeric TRIM25 RBCC has a cross-sectional radius of 31 Å, which is very similar to that of the tetrameric TRIM32 RBCC (32 Å). These observations suggest a model for the structure of the TRIM25 RBCC fragment in which the RING and B-box domains might fold back across the coiled-coil in a fashion similar to that observed in the crystal structure of the TRIM5α BCC fragment, where each B-box, on either side of the CC stem, packs back onto the helices (Fig EV3) (Goldstone *et al*, 2014). In contrast, the TRIM32 RBCC construct is the longest with a $D_{max}$ of 243 Å and a particle central diameter of ~56 Å that can easily accommodate the width of 2 coiled-coil stems (~18 Å from the structure of TRIM5α). Two lobes are present on each side of the envelope and can accommodate two RB2 domains. The analysis of the SAXS data indicates that TRIM32 has a structure that is consistent with a canonical anti-parallel long coiled-coil domain that places the B-box and RING domains at opposite ends of the CC with the RING being constitutively dimerized, thereby locking the structure at both ends into a stable tetrameric form (Figs 7 and EV4).

## Discussion

This study was aimed at gaining insight into how self-association of TRIM ligases regulates their function and how individual

domains of the tripartite motif contribute to the observed oligomeric state and associated catalytic activity. Our data uncover that different members of the TRIM family use diverse mechanisms for self-association that is likely linked to their specific biological functions. However, our data also highlight that the structural arrangements used to activate the E2~ubiquitin intermediate have common features between different family members. These involve dimerization of the RING to maintain a closed and hence active E2~Ub conformation and interactions between a conserved acidic residue outside the core RING to further stabilize the active state.

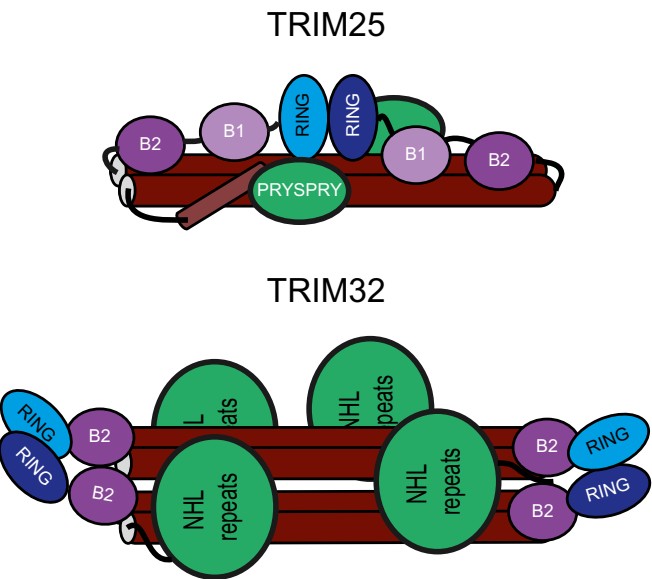

**Figure 7.  Model of RING dimerization in different TRIM ligases.**
TRIM25 and TRIM32 use different mechanisms to allow their respective RING domains to dimerize: in TRIM25, the propensity to dimerize is low and happens in an intramolecular fashion possibly aided by E2~Ub and substrate binding. It is possible that the B-box domains may contact the CC region to support dimerization. The RINGs of TRIM32 are constitutive dimers even in the absence of the rest of the remainder of the protein or additional binding partners. Tetramerization of TRIM32 may be important for the recognition of low-affinity substrates or may allow co-localization of different binding partners.

The RING domains of both, TRIM25 and TRIM32, crystallize as dimers, in both cases with dimerization mediated by helices on either side of the RING core. Importantly, the structure of the TRIM25 RING/E2~Ub complex shows how dimerization is used to enhance the rate of ubiquitin transfer: by stabilization of a closed E2~Ub conformation as observed previously for the dimeric RINGs RNF4 and BIRC7 (Dou *et al*, 2012; Plechanovova *et al*, 2012). In addition, our structure highlights that TRIM25 uses an additional feature to enforce the closed conformation that is reminiscent of the monomeric E3 Cbl-b that uses a phospho-tyrosine outside the RING domain to stabilize the E2~ubiquitin intermediate (Dou *et al*, 2013). There are no reports of phosphorylation events adjacent to the RING domain regulating TRIM catalytic activity, and instead, TRIM25 uses a glutamate in the N-terminal helix to contact the opposite ubiquitin in a spatial arrangement that is highly similar to the pTyr of Cbl-b. Interestingly, this Glu is conserved in many TRIM ligases, including TRIM32 (Li *et al*, 2014), and mutation to Arg severely disrupts catalytic activity in either protein, suggesting that its functional role is conserved. At present, it is not clear why TRIM ligases would require two different structural elements to stabilize the closed E2~Ub conformation but we speculate that it could either provide an additional level of regulation or may be used to further stabilize the closed conformation in cases where RING dimerization is weak or when the specific orientation of the RINGs in the dimer may prevent efficient contact of both RINGs with ubiquitin.

In TRIM25, dimerization of the RING occurs with very low affinity in solution, and even at concentrations around 100 μM, no significant amount of dimer can be detected. However, the complex between the RING and the E2-Ub conjugate crystallized as a dimer suggesting that binding to the ubiquitin–loaded E2 stabilizes the otherwise weak RING–RING interaction. Such an increase in apparent affinity most likely explains why we can detect catalytic activity under assay conditions in which the RING is largely monomeric.

The B-boxes of TRIM25 and TRIM32 have no apparent effect on oligomerization and only a small effect on catalytic activity, clearly setting them apart from the best studied member of the TRIM family, the retroviral restriction factor TRIM5α, in which the B-box promotes higher order oligomerization (Diaz-Griffero *et al*, 2009; Li *et al*, 2011). This raises questions about the role of B-boxes in those TRIMs that do not require additional oligomerization domains.

A recent kinetic study supports a model in which TRIM32, TRIM25 and TRIM5α catalyse poly-ubiquitin chain formation through a cooperative allosteric mechanism, which would explain the requirement of oligomerization for catalytic activity (Streich *et al*, 2013). However, TRIM5α showed the lowest Hill coefficient in that study, which is intriguing given that TRIM5α forms higher order oligomers in contrast to the dimeric and tetrameric forms of TRIM25 and TRIM32, respectively. The molecular basis for this observation is not clear at present but may be due to differences in experimental conditions, and it will be interesting to see in future studies how the observed kinetic behaviour is linked to the structure and mechanism of these ligases.

Interestingly, *ab initio* models for envelopes of the RBCC regions of TRIM25 and TRIM32 indicate that tetrameric TRIM32 is more elongated despite containing only one B-box but has a diameter similar to the TRIM25 RBCC dimer (Figs 6 and EV3). This suggests a model in which the RING and B-box domains of TRIM25 may bind back across the coiled-coil, which in turn could allow the two RING domains to

interact in an intramolecular fashion. Such a model is supported by the observation that full-length TRIM25 is dimeric, and hence, the only mechanism to form a dimeric RING is in an intramolecular manner (Li *et al*, 2014). Furthermore, it has recently been shown that the C-terminal PRYSPRY domain of TRIM20 folds back across the coiled-coil (Weinert *et al*, 2015). If the PRYSPRY of TRIM25 behaved in a similar fashion, this could result in an overall architecture in which the RING and substrate-binding PRYSPRY domains are located in close proximity along the coiled-coil to facilitate ubiquitin transfer from the E2 onto the substrate (Figs 7 and EV4). In such an arrangement, RING dimerization could potentially be enhanced by substrate binding, thereby ensuring that TRIM25 activity is only available in the correct setting. This is particularly interesting in the light of the observation that viral proteins such as NS1 bind to the coiled-coil region of TRIM25, which could prevent intramolecular RING dimerization in this model and hence activity (Gack *et al*, 2009; Rajsbaum *et al*, 2012). In contrast, in TRIM32 the RING acts as an independent dimerization module and promotes formation of a TRIM tetramer, in which the RING dimers are located on either side of the central coiled-coil (Figs 7 and EV4). Such an arrangement provides four substrate-recognition domains, which could be important to enhance substrate binding through avidity effects or may even bring different substrates together. Further studies are now required to test these models, but here, we have highlighted an unexpected diversity in the mechanism TRIM ligases use to couple oligomerization to catalytic activity and substrate recognition with more variations on this theme likely to be discovered in the future.

## Materials and Methods

### Protein production and purification

Cloning, expression and purification of His-Ube1, UBE2D1, UBE2D3, UBE2N, UBE2V1, His$_6$-M1C-ubiquitin and mutants have been described before (Carvalho *et al*, 2012; Stieglitz *et al*, 2012). TRIM25 constructs RING (1–82), RB1 (1–152), RB1B2 (1–202), RBCC (1–433) and mutants present in this study were cloned into a modified pET-52b with a SUMO-tag to produce cleavable His$_6$-SUMO fusion proteins. TRIM32 RING constructs (1–93, 7–93 and 10–84) were expressed with a C-terminal His$_6$-tag in pET28 and TRIM32 RB2 (1–140) and RBCC (1–354) in pET49b with a cleavable N-terminal GST-tag. All proteins were expressed in BL21 (DE3) *E. coli* cells. Media were supplemented with 100 μM ZnCl$_2$, and cells were induced at 0.6 OD$_{600}$ with 150 μM isopropyl β-D-1-thiogalactopyranoside (IPTG) and incubated at 16°C for 16 h. Protein was purified by affinity chromatography, followed by ion-exchange chromatography (after removal of the His-SUMO or GST-tag by HRV-3C) and size-exclusion chromatography (SEC). His$_6$-M1C-ubiquitin was labelled with Atto 647N maleimide (Sigma) as described for Cy5 labelling (Stieglitz *et al*, 2012). All plasmids were verified by DNA sequencing. Protein molecular mass was verified by electrospray ionization mass spectrometry. The fold of proteins was analysed by circular dichroism spectroscopy. Protein concentrations were determined by UV absorption at 280 nm using calculated extinction coefficients or Bradford assays for constructs that do not contain Trp or Tyr residues. Bovine mono-ubiquitin was purchased from Sigma and further purified by SEC.

## Production of the UBE2D~Ub thioesters and isopeptide-linked UBE2D1-Ub

The UBE2D~Ub intermediates were synthesized as previously described (Stieglitz *et al*, 2013). Briefly, His-Ube1 (1 μM), UBE2D3 or UBE2D1 (250 μM), Ub or Atto 647N-Ub (Ub$^{Atto}$) (500 μM) and ATP (3 mM) (Sigma) were incubated in reaction buffer containing 50 mM HEPES pH 7.5, 150 mM NaCl, 20 mM MgCl$_2$ for 60 min at 25°C. The UBE2D~Ub thioester-linked intermediate was purified by SEC using a HiLoad 16/60 Sephadex 75 gel filtration column (GE Healthcare) pre-equilibrated in 50 mM HEPES pH 7.5, 150 mM NaCl. Similarly, the isopeptide-linked UBE2D1-Ub used in structural studies was prepared according to Plechanovova *et al* (2012). His-Ube1 (1 μM), UBE2D1 (S22R/C85K) (200 μM), Ub (300 μM) and ATP (3 mM) were incubated in reaction buffer containing 50 mM Tris pH 10, 150 mM NaCl, 20 mM MgCl$_2$ for 16 h at 30°C and subsequently purified by SEC as above.

## *In vitro* ubiquitination and ubiquitin discharge assays

Ubiquitin discharge assays with pre-charged UBE2D~Ub were performed with 10 μM UBE2D3~Ub adduct and 4 μM TRIM construct in buffer containing 50 mM HEPES pH 7.5, 150 mM NaCl and 50 mM L-lysine for InstantBlue (Expedeon) stained gels or 1 μM UBE2D1~Ub$^{Atto}$ adduct, 4 μM TRIM construct and 20 mM L-lysine for quantification by fluorescence detection. Reactions were incubated at 25°C for up to 30 min, and samples were quenched with 2× SDS sample buffer at the described time intervals and resolved by SDS–PAGE. For quantification, gels were scanned with a Storm 869 Scanner and the bands for E2~Ub$^{Atto}$ and Ub$^{Atto}$ integrated using ImageQuant (both GE Healthcare). It was not possible to use the ratio of E2~Ub$^{Atto}$/Ub$^{Atto}$ as a readout as a small portion of ubiquitin was transferred to the E3, especially for longer constructs (Fig EV1). Experiments were performed in duplicate (mutants) or triplicate.

Ubiquitination assays with UBE2N/UBE2V1 were performed with 1 μM Ube1, 10 μM each E2, 4 μM TRIM construct and 100 μM ubiquitin in buffer containing 50 mM HEPES pH 7.5, 150 mM NaCl, 20 mM MgCl$_2$ and 10 mM ATP and visualized by InstantBlue staining. For quantification, assays were carried out with 0.5 μM Ube1, 2.5 μM of each E2, 4 μM TRIM construct and 50 μM ubiquitin, supplemented with 1 μM Ub$^{Atto}$. Gels were scanned and the bands for free Ub$^{Atto}$ integrated. Experiments were performed in triplicate.

## SEC-MALLS

Analytical SEC-coupled with multi-angle laser light scattering (MALLS) profiles were recorded at 16 angles using a DAWN-HELEOS-II laser photometer (Wyatt Technology) and differential refractometer (Optilab TrEX) equipped with a Peltier temperature-regulated flow cell maintained at 25°C (Wyatt Technology). Samples of purified proteins at different concentrations (100 μl volume) were applied to a Superdex 75 (for smaller constructs) or Superdex 200 (for larger TRIM constructs) 10/300 GL column (GE Healthcare) equilibrated with 20 mM HEPES pH 7.5, 100 mM NaCl (TRIM32) or 150 mM NaCl (TRIM25), 0.5 mM TCEP and 3 mM NaN$_3$ at a flow rate of 0.5 ml/min. The data were analysed using ASTRA 6.1.

## SAXS data collection and analysis

Synchrotron SAXS data were collected at the Diamond Light Source on beamline B21 and the SWING beamline at SOLEIL. The Diamond data were recorded on a Pilatus 2M detector with a fixed camera length of 3.9 m and 12.4 keV energy allowing the collection of a momentum transfer range, $q$ between 0.015 and 0.3 Å$^{-1}$. All samples were extensively dialysed against the background buffer and measured for at least three construct dilutions in the concentration range that gave a monodisperse molecular weight according to SEC-MALLS. The data at SOLEIL were recorded over a momentum transfer range of 0.01–0.43 Å$^{-1}$. The two purified TRIM RBCC constructs (2.5 mg/ml TRIM25 and 6 mg/ml TRIM32) were injected onto an SEC-3 300 Å Agilent column and eluted at a flow rate of 0.2 mg/ml at 15°C. Frames were collected continuously during the fractionation of the proteins. Frames collected before the void volume were averaged and subtracted from the signal of the elution profile to account for background scattering. Data reduction, subtraction and averaging were performed using the software FOXTROT (SOLEIL).

The scattering curves were analysed using the programs Scatter and the package ATSAS to obtain the radius of gyration ($R_g$), the maximum particle dimension ($D_{max}$), the excluded particle volume ($V_p$), the cross-sectional radius and the pair distribution function (P(r)) (Petoukhov *et al*, 2012). The molecular mass of the scattering particles was estimated using a method described by Rambo (Rambo & Tainer, 2013). Low-resolution three-dimensional *ab initio* models for the different constructs of TRIM25 and 32 (with applied P2 symmetry for the dimeric and tetrameric constructs) were generated by program DAMMIF, averaging the results of 25 independent runs using the programs SUPCOMB and DAMAVER (Franke & Svergun, 2009). The SAXS-derived dummy atoms models were rendered with PyMOL (Schrödinger, LLC).

## NMR spectroscopy

All spectra were recorded at 25°C on Bruker AVANCE spectrometers operating at 14.1 T and 16.5 T in NMR buffer (20 mM HEPES, pH 7.5, 100 mM NaCl, 0.5 mM TCEP). Data were processed with NMRPipe/NMRDRAW and analysed with CCPN software (Delaglio *et al*, 1995; Vranken *et al*, 2005). $^{15}$N relaxation measurements and $^1$H-$^{15}$N heteronuclear NOE were collected as previously described (Kay *et al*, 1989). R1 and R2 values were determined for each residue by fitting an exponential decay to the peak intensity of data collected in an interleaved manner to minimize time-dependent temperature or stability effects with delay times in random sequence. T1 longitudinal recovery delays were set to 10, 100, 200, 400, 600, 800, 1,200 and 1,600 ms. T2 transverse recovery delays were set to 8, 16, 24, 40, 56, 80, 104 and 128 ms. In each case, the error was determined from the fit according to a procedure implemented in CCPN analysis. Residues were excluded in which overlap in the data precluded accurate measurement of the peak intensity and where the value of heteronuclear NOE (calculated from the ratio [Peak Intensity$_{saturated}$]/[Peak Instensity$_{unsaturated}$]), indicative of local motion or chemical exchange, was below 0.7. Isotropic correlation times were determined using the program TENSOR2 (Dosset *et al*, 2000). The uncertainties on the values of the isotropic correlation times are obtained by a Monte-Carlo simulation procedure implemented in TENSOR2, which assumes that the stochastic fluctuations of the measured relaxation rates are described by their experimental

errors. Isotropic correlation time projections were obtained by the program HYDRONMR (Garcia de la Torre *et al*, 2000).

### Crystallization, data collection, phasing and refinement

TRIM25 RING domain (8 mg/ml) was mixed with UBE2D1-Ub (10 mg/ml) at a 1:1 molar ratio and crystallization trials set up using an Oryx crystallization robot. Initial hits were optimized by hanging-drop vapour diffusion at 18°C with a reservoir solution containing 100 mM Tris pH 8.5 and 10% PEG 20,000 reaching full size (0.5 mm) after 3 days. Crystals were flash-frozen in the reservoir solution containing 20% glycerol. TRIM32 RING domain crystals were grown using an Oryx crystallization robot in drops consisting 0.3 μl of protein at a concentration of ~6 mg/ml and 0.3 μl of reservoir solution (0.1 M MES pH 6.0 and 5% PEG 600). Crystals appeared overnight and were flash-frozen in the reservoir buffer with 35% PEG 600.

### Data collection and structure determination

Data for TRIM25R:UBE2D1-Ub crystals were collected on beamline IO4 ($\lambda = 1.2829$ Å) at the Diamond Light Source (Oxford, UK) and processed using XDS. Data for TRIM32 RING domain crystals were collected on beamline ID29 at the ESRF Synchrotron Radiation Facility (Grenoble, France) at a wavelength of 1.28254 Å and processed using XDS (Kabsch, 2010). Both structures were solved by single-wavelength anomalous dispersion phasing using the $Zn^{2+}$ atom. Heavy-atom search, density modification and initial model building were performed using Phenix AutoSol (Adams *et al*, 2010). Models were iteratively improved by manual building in Coot and refined using REFMAC5 and Phenix (Murshudov *et al*, 1997; Emsley & Cowtan, 2004). The stereochemistry of the final models was analysed with Procheck. There is no electron density for amino acids 55–58 in both TRIM32 monomers and amino acids 70–74 in one monomer. All structural figures were prepared in PyMOL.

Coordinates and structure factors were deposited in the Protein Data Bank under accession codes 5FER and 5FEY.

**Expanded View** for this article is available online.

### Acknowledgements

We thank L. Haire and R. Ogrodowicz for help with crystallization, L. Masino for circular dichroism spectroscopy, P. Walker for data collection, S. Smerdon for discussions and advice, the MRC Biomedical NMR Centre for access and advice and the Diamond Light Source, Oxford, ESRF Synchrotron Radiation facility, Grenoble and Soleil, Gif-sur-Yvette for synchrotron access and group members for advice and helpful discussions. This work was supported by the Medical Research Council (grant U117565398) until March 2015 and from April 2015 by the Francis Crick Institute (grant number FCI01) which receives its core funding from Cancer Research UK, the UK Medical Research Council and the Wellcome Trust. In addition, this work was supported by a PhD fellowship from the Boehringer Ingelheim Fonds to M.G.K.

### Author contributions

MGK and DE designed and carried out experiments, analysed the data and co-wrote the manuscript. EC helped with cloning and protein expression, IAT helped with MALLS and analysed and discussed the data, KR conceived the project, analysed the data and wrote the manuscript.

### Conflict of interest

The authors declare that they have no conflict of interest.

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
