## [Review Process File · The EMBO Journal]

Manuscript EMBO-2015-93741

Functional role of TRIM E3 ligase oligomerization and regulation of catalytic activity

Marios G. Koliopoulos, Diego Esposito, Evangelos Christodoulou, Ian A. Taylor and Katrin Rittinger

Corresponding author: Katrin Rittinger, The Francis Crick Institute

Review timeline:

Submission date:	17 December 2015
Editorial Decision:	26 January 2016
Revision received:	10 March 2016
Editorial Decision:	30 March 2016
Revision received:	01 April 2016
Accepted:	06 April 2016

Transaction Report:

Editor: Hartmut Vodermaier

1st Editorial Decision

26 January 2016

Thank you again for submitting your manuscript on TRIM ligases for our consideration. Following some turn-of-the-years-related delays, for which I apologize, we have now received a complete set of reviews on your manuscript. All three referees acknowledge the potential interest of this work, yet remain somewhat ambiguous regarding suitability of the study for The EMBO Journal at the present stage. As you will see from the comments copied below, their main criticisms are less related to major technical problems, but derive from important conceptual concerns regarding the rationale and setup of the study, the interpretation and discussion of the results, and the significance of this work in the broader context of both biology and biochemistry of TRIM proteins. Given that these reservations are intimately linked to the overall presentation of the study, I feel that the referees might well become more enthusiastic and supportive following major reorganization and rewriting of the manuscript, and I would therefore like to give you an opportunity to revise the study along the lines suggested in their comments; but the final decision regarding publication in our journal will depend on the referees' opinion on the revised manuscript and point-by-point response. I should also point out that in addition to the major presentational changes, the various technical issues raised by the reviewers should be addressed as well through additional experimental analyses - including the concerns regarding Ubc13 usage, inadequacy of single discharge assays in some experiments, and the missing quantification of discharge and chain formation assays.

I realize that these modifications may require significant additional time and effort, but in light of the referee reports feel that they would nevertheless appear essential in order to make this study a highly compelling candidate for EMBO Journal publication. In any case, our scooping protection policy means that we would not take competing studies appearing elsewhere during the revision period (standard three months, but extendable up to six months upon request) into account for our final decision on your study.

Should you have any questions or comments regarding the referee reports and this decision, please do not hesitate to contact me directly. Thank you again for the opportunity to consider your work for publication, and I look forward to your revision.

REFeree REPORTS:

Referee #1:

The Trim ligases constitute a superfamily of over 100 human paralogs possessing a highly conserved domain architecture. Of the Trim ligases functionally characterized to date, many are highly pleiotropic in regulating cellular processes and in participating in the innate immune response of the cell. They are all characterized by the presence of an N-terminal Ring catalytic domain that recognizes a cognate E2~ubiquitin thioester activated intermediate, generated by ubiquitin activating enzyme (E1), one or two B box domains and an extended coiled coil oligomerization region, after which may be appended additional protein interaction domain(s) for recruiting target proteins substrates. While structures for selected regions of the proteins have been determined in the past, how the overall structure is organized to allow substrate targeting and polyubiquitin chain assembly remains elusive. Several groups, cited in the manuscript, have determined structures for the coiled coil region and determined that oligomerization occurs with antiparallel orientation of the Ring catalytic "head" domains. The present manuscript fills in this gap by providing structures for the orientation of the "head" regions for two Trim orthologs having different B domain topologies. The structural data emphasizes the difference in overall folding for otherwise similar domain architectures that obtain with the end results of juxtaposing the Ring domains into a catalytic ensemble capable of binding E2~ubiquitin thioester into the "activated" conformer. The authors argue that oligomerization is thus required for catalytic activity. Structural analysis of nested truncations by light scattering is an important contribution to confirming the importance of the coiled coil region in oligomerization and in defining the stoichiometry of the oligomers. The crystal structures are important in defining the orientation of the dimeric Ring domains with bound E2~ubiquitin thioester.

The requirement of oligomerization for catalytic activity is not a new concept, as suggested by the authors, but was first shown in kinetic studies of Trim32, Trim25, and Trim5alpha by Streich et al. [J. Biol. Chem. 288, 8209-8221 (2013)], of which it is surprising the authors are unaware. In fact, the current observations should be considered as confirming the empirical predictions of the earlier kinetic work. Streich et al. demonstrated that the three Trim ligases showed identical specificity for a single E2 family (Ubc5) and that the enzymes exhibited allosteric cooperativity in assembly of polyubiquitin chains, thus requiring oligomerization. Indeed, the Hill coefficients (defining the lower limit to number of interacting sites) are consistent with the Ring domain stoichiometries shown in the present structural work, especially if one considers two E2~ubiquitin binding sites per Ring domain, as proposed by Streich et al. In addition, the multiple Ring domain arrangement reconciles the antiparallel orientation of the "head" domains with the kinetic data (viz a viz, the Hill coefficients) and the differences in Hill coefficients between Trim32 and Trim25. Minimally, the authors need to cite this earlier kinetic work and how the present structural work corroborates and extends the empirical predictions. Better still, the authors should consider the earlier data and reconcile their structure with the kinetic data in considering in a broader sense how this Ring architecture informs on the mechanism of chain formation.

Other points:

1. General comment- I strongly urge that, for clarity, the authors not refer to the thioester-linked E2~ubiquitin intermediates as "conjugates." While semantically correct, in the ubiquitin field

"conjugates" have an historically-special meaning of referring to isopeptide-linked ubiquitin-protein adducts. While I understand that some investigators in the field continue to refer to thioesters as "conjugates," use of the term adds nothing to understand these intermediates.

2. Page 9, lower half of page- The discharge assays are too limited and qualitative to be used for these studies since it measures only formation of the closed "activated" E2~ubiquitin conformer. Therefore, it is not surprising that very similar results are being observed here. Far more useful would be using polyubiquitin chain formation as the functional readout.

3. Page 10, line 5- I am unaware of any published work claiming Ubc13 supports the Trim activity of the enzymes studied here. If the authors wish to continue to include this claim then they should provide a citation of the relevant paper. It is unclear if Ubc13 is being included as a control or as a bona fide cognate E2 in these studies; indeed, the complete lack of activity with Ubc13 is inconsistent with it functioning to support the Trim ligases and would support the earlier conclusion that only Ubc5 isozymes are the cognate E2s for these enzymes.

4. Page 10, first full paragraph- The data do not "indicate" but merely confirm that the Ring domain is the main catalytic unit, as this was shown by Streich et al. and subsequently others. Other conclusions in this paragraph are based on the likely spurious Ubc13 results.

5. Page 12, line 9, "Given that no dimer..."- This conclusion is unsupported by the data since it is based on negative biophysical data. The next sentence is incorrect because it is based on the irrelevant Ubc13 paralog.

6. Page 15, line 5 from bottom- "by helices (on) either side..."

7. Page 17, line 8 from bottom- "...are located (on) either side..."

8. Page 18, line 6 from bottom; page 19, line 7- "MgCl₂" needs 2 as subscript, not superscript

Referee #2:

The authors describe a focused biophysical and biochemical study of the role of oligomerization on the catalytic activity of TRIM ubiquitin ligases. TRIMs contain dimerization domains in the form of coiled-coils and are therefore often assumed require dimerisation for full activity. The authors report solution-based analyses of associations of various TRIM constructs, catalytic activity of the constructs as measured by both discharge from an E2 conjugating enzyme and in the formation of ubiquitin chains linked across lysine 63. They also determine the structures of one TRIM (32) - the solution structure of the RING of TRIM 32 is a monomer, but addition of 10 residues to the N-terminus and 9 residues to the C-terminus renders it dimeric, and this structure is crystallized. They also solve the structure of the TRIM25 RING domain in complex with a ubiquitin-bearing Ube2D1 species. This RING remains monomeric in solution but also forms a similar dimer in crystals.

Overall, the data are of high quality, the manuscript is well-written. One weakness is that there is not much discussion of the biology of the TRIMs - a short sentence in the introduction regarding CARDS (to the non-specialist, this is a little difficult to follow).

I have some minor concerns that the authors could address prior to publication. In some places the writing is a bit casual - 'it is believed' comes up a lot and I'm not sure I understand the meaning. For example the statement that self-association via coiled-coil is believed to be crucial for activity - what evidence is this belief based on? Has it been reported to be crucial, or put forward as a hypothesis? Similarly B-boxes are believed to act as PPI motifs.

The title of the section small angle x-ray scattering could be changed to indicate the findings in figure 2 rather than the technique used.

In the comparisons of activity in figure 3 (and 5A) the different activity levels might be easier to see in a graphical format, some quantification and statistical analysis. For example the ratio of E2~ub to E2 can be used to define the 'activity' of the TRIM constructs. This would allow the representation not to include the TRIM construct itself as currently sit, asterisked which draws the attention to this band on the gels. Also, it is a matter of preference but there are efforts in the ubiquitin fields to standardize the names of the E2s, so including Ube2D3 where UbcH5c is used, and Ube2D1 where UbcH5A is used, Ube2N in place of Ubc13, UBE2V1 would be helpful since many researchers use this nomenclature.

Also - is there a rationale for using one Ube2D family member for assays and a different one for crystallization? The section in methods detailing production needs careful proofing to ensure each enzyme is referred to correctly.

In figure 4B I found the color differences between TRIM32 and TRIM 25 difficult to follow because blue/cyan is used in figure 4A and 4C to denote monomers within one dimer.

Fig 5C is referred to in the text before 5A and 5B are. Perhaps the order of the panels in figure 5 needs rethinking.

The sequence limits for each construct would be useful in the schematic in figure 1.

p3. that often contains protein interaction motifs should read protein-protein interaction motifs.

When describing RBRs as catalytic there should be reference to Wenzel (2011) Nature .

Guinier is mis-spelled as gunier in the text and in table 1.

Referee #3:

This manuscript investigates the architecture and activity of two TRIM E3 ligases - TRIM25 and TRIM32. The authors utilize a number of different constructs from each protein in a range of biochemical and structural studies. The key conclusions/observations in the manuscript are: (i) that irrespective of the oligomeric state of the RING domain in solution it appears that RING dimerization is required for ubiquitin transfer; (ii) that RING dimerisation may be stabilized by binding the E2~ub conjugate; (iii) The RING dimer traps the E2~Ub conjugate in a closed conformation that is comparable to other RINGs, however, electrostatic interactions - that resemble those between Cbl-b and the conjugated ubiquitin, are important; (iv) In the absence of E2~Ub conjugate the oligomeric arrangement of TRIMs varies, as does the impact of additional domains - i.e. in the absence of the coiled coil domain TRIM25 is a monomer, but TRIM32 is a dimer; (v) A model of how the anti-parallel coiled-coil domain of the TRIMs allows RING dimerization in the presence and absence of a higher order structure - as represented by TRIM32 and TRIM 25 respectively, is proposed.

TRIM proteins are a large family of E3 ligases that are reported to influence key signaling events in cells and this work will be of considerable interest to the wider field. The data is mostly of high quality and the conclusions well supported, but the manuscript is not easy to read and the authors should consider revising it to make the material more accessible to a wider audience. For example, it may be easier for the reader if the crystal structure of the complex is presented first, followed by the mutagenesis and then the solution SAXS studies, which together lead to a model of assembly and activity. As a minimal requirement the SAXS and NMR sections should be revised and may be better presented together.

Major points:

1. As indicated above the SAXS data might be more meaningfully discussed after the structure solution. As presented it is a very dense read for a non-specialist, general audience as the

significance of the data is not necessarily apparent at this stage

2. The assays for RING activity used throughout the paper measure either hydrolysis of the thioester bond linking the ubiquitin to the E2, UbcH5c, or formation of polyubiquitin chains by Ubc13. None of the assays are quantified - as the effect of domain deletion/mutation is sometimes modest some assays should be quantified. For Ubc13, activity is indicated as being comparable in Figure 3b but there is some indication of increased processivity for T25-RBCC - and given subsequent conclusions about dimerization/activity, this may warrant closer inspection.

3. In figure 5 it would be interesting to analyse the V72R mutant in the linked dimer. Also the structure presented in panel 5C might be better shown first.

4. The authors suggest that addition of the E2~Ub conjugate triggers RING dimerization of TRIM25 RING. Can this be observed using MALS analysis?

5. Further, in figure 5 enhanced discharge was observed with the fused RINGs but chain building was not altered - does this reflect differences in the ability of the different conjugates to stabilize the RING dimer? If this is not the reason what other differences might account for the data.

6. In general, consider reorganizing the order data appears in the manuscript so that the model presented in the discussion is better developed based on the data.

Minor points:

- 1) For TRIM25 use either RBCC or RBBCC, but not both for the long construct.
- 2) The use of TRIM25 and TRIM32 in many figures/panels has the potential to confuse the reader. Wherever possible the figures should indicate the protein studied - ie. The MALS data in Figure 1b could be labeled with the relevant TRIM.
- 3) Figure 4b - TRIM 32 or TRIM37? Also include relevant pdb identifiers in the text.
- 4) It would be helpful to include the sequence of the crystallised TRIM domains in the extended data.

1st Revision - authors' response

10 March 2016

Point by point response, EMBOJ-2015-93741

“Functional role of TRIM E3 ligase oligomerization and regulation of catalytic activity“

We would like to thank the reviewers for their positive and constructive comments. We have addressed all the issues raised as detailed below. Specifically, we have performed additional experiments to quantify activity assays, have restructured the manuscript as requested and have further emphasized the studies that describe that UBE2N/Ubc13 is a bona fide E2 for TRIM25 and TRIM32. Due to the additional experiments and changes requested we made a number of changes to the figures as described in detail in our point by point response.

We believe these changes have significantly improved our manuscript and we hope that the reviewers will now find the revised manuscript suitable for publication.

Referee #1:

The Trim ligases constitute a superfamily of over 100 human paralogs possessing a highly conserved domain architecture. Of the Trim ligases functionally characterized to date, many are highly pleiotropic in regulating cellular processes and in participating in the innate immune response of the cell. They are all characterized by the presence of an N-terminal Ring catalytic

domain that recognizes a cognate E2~ubiquitin thioester activated intermediate, generated by ubiquitin activating enzyme (E1), one or two B box domains and an extended coiled coil oligomerization region, after which may be appended additional protein interaction domain(s) for recruiting target proteins substrates. While structures for selected regions of the proteins have been determined in the past, how the overall structure is organized to allow substrate targeting and polyubiquitin chain assembly remains elusive. Several groups, cited in the manuscript, have determined structures for the coiled coil region and determined that oligomerization occurs with antiparallel orientation of the Ring catalytic "head" domains. The present manuscript fills in this gap by providing structures for the orientation of the "head" regions for two Trim orthologs having different B domain topologies. The structural data emphasizes the difference in overall folding for otherwise similar domain architectures that obtain with the end results of juxtaposing the Ring domains into a catalytic ensemble capable of binding E2~ubiquitin thioester into the "activated" conformer. The authors argue that oligomerization is thus required for catalytic activity. Structural analysis of nested truncations by light scattering is an important contribution to confirming the importance of the coiled coil region in oligomerization and in defining the stoichiometry of the oligomers. The crystal structures are important in defining the orientation of the dimeric Ring domains with bound E2~ubiquitin thioester.

The requirement of oligomerization for catalytic activity is not a new concept, as suggested by the authors, but was first shown in kinetic studies of Trim32, Trim25, and Trim5alpha by Streich et al. [*J. Biol. Chem.* 288, 8209-8221 (2013)], of which it is surprising the authors are unaware. In fact, the current observations should be considered as confirming the empirical predictions of the earlier kinetic work. Streich et al. demonstrated that the three Trim ligases showed identical specificity for a single E2 family (Ubc5) and that the enzymes exhibited allosteric cooperativity in assembly of polyubiquitin chains, thus requiring oligomerization. Indeed, the Hill coefficients (defining the lower limit to number of interacting sites) are consistent with the Ring domain stoichiometries shown in the present structural work, especially if one considers two E2~ubiquitin binding sites per Ring domain, as proposed by Streich et al. In addition, the multiple Ring domain arrangement reconciles the antiparallel orientation of the "head" domains with the kinetic data (viz a viz, the Hill coefficients) and the differences in Hill coefficients between Trim32 and Trim25. Minimally, the authors need to cite this earlier kinetic work and how the present structural work corroborates and extends the empirical predictions. Better still, the authors should consider the earlier data and reconcile their structure with the kinetic data in considering in a broader sense how this Ring architecture informs on the mechanism of chain formation.

We very much apologise for not referencing the manuscript by Streich and colleagues, 2013. This was a regrettable oversight and this work is now being referenced in the Introduction and further discussed in Discussion.

It was never our intention to imply that the requirement of oligomerisation for catalytic activity is a new concept invented by us. The message we tried to convey was that while oligomerisation as a functional necessity for catalytic activity has been suggested before, a systematic analysis of the link between oligomerisation and catalytic activity has been missing so far. We have rephrased the text accordingly.

We are now commenting in the Discussion on the differences in Hill coefficients determined by Streich and colleagues which are $n=5.1$ (TRIM32), $n=3.6$ (TRIM25) and $n=2.9$ (TRIM5 α). However, it is very difficult to discuss these values in depth as our MALLS-derived TRIM stoichiometries predict 2 E2 binding sites for dimeric TRIM25 and 4 E2 binding sites for tetrameric TRIM32. The situation in the case of TRIM5 α is even more complicated given that this TRIM member has been shown in many studies to form higher order oligomers (dimerisation via the CC domain and further association via the B-box) and hence it is not clear how this may fit with an n of 2.9.

Other points:

1. General comment- I strongly urge that, for clarity, the authors not refer to the thioester-linked E2~ubiquitin intermediates as "conjugates." While semantically correct, in the ubiquitin field "conjugates" have an historically-special meaning of referring to isopeptide-linked ubiquitin-protein adducts. While I understand that some investigators in the field continue to refer to thioesters as "conjugates," use of the term adds nothing to understand these intermediates.

We used the term “conjugates” as many researchers in the ubiquitin field use this expression. However, for clarity we have now changed “conjugate” to “intermediate” throughout the manuscript.

2. Page 9, lower half of page- The discharge assays are too limited and qualitative to be used for these studies since it measures only formation of the closed "activated" E2~ubiquitin conformer. Therefore, it is not surprising that very similar results are being observed here. Far more useful would be using polyubiquitin chain formation as the functional readout.

This reviewer correctly points out that E2~Ub discharge assays are a measure of how efficiently a given ligase stabilises the closed activated state of the E2~Ub intermediate. We chose this experimental set-up as poly-ubiquitin chain formation assays conducted with UbcH5 (UBE2D) isoforms result primarily in autoubiquitination of the TRIM constructs used. The TRIM constructs that are compared in these assays are of different length and therefore contain vastly different numbers of lysine residues and hence the apparent activity detected in ubiquitination assays with UbcH5 isoforms is in large part a measure of the number of lysine residues available for modification (explained on page 7 of the manuscript).

Discharge assays are therefore a much cleaner readout to follow the ability of a given ligase construct to stabilise the closed, active E2~Ub form.

These discharge assays have now been quantified using a fluorescently-labelled ubiquitin as also requested by the other reviewers (see Figures 2, 4, 5, EV1 and EV2). The use of a fluorescently-labelled ubiquitin in these assays was necessary as some of the TRIM constructs analysed have a similar MW as the other assay components and hence their bands overlap on SDS gels preventing quantification of Coomassie-stained gels. For further details of these new assays please see our reply to reviewer 2.

Nevertheless, we fully agree that it is important to also follow poly-ubiquitin chain formation as a functional readout and therefore we had carried out assays with the Ubc13/Uev1a (UBE2N/UBE2V1) heterodimer. This heterodimeric E2 synthesises unattached K63-linked poly-ubiquitin chains and hence the activity readout is not influenced by the length of the ligase construct under investigation (data presented in Figures 2, 4 and 5). The physiological relevance of using this E2 to follow the activity of TRIM25 and TRIM32 is explained in detail in our response to point 3.

3. Page 10, line 5- I am unaware of any published work claiming Ubc13 supports the Trim activity of the enzymes studied here. If the authors wish to continue to include this claim then they should provide a citation of the relevant paper. It is unclear if Ubc13 is being included as a control or as a bona fide cognate E2 in these studies; indeed, the complete lack of activity with Ubc13 is inconsistent with it functioning to support the Trim ligases and would support the earlier conclusion that only Ubc5 isozymes are the cognate E2s for these enzymes.

We are surprised by this statement and believe it must be a misunderstanding. As described in the introduction on page 5, both ligases, TRIM25 and TRIM32 have been shown to work with Ubc13 as described by James Chen's group in 2010 (Zen et al.), Germana Meroni's group in 2011 (Napolitano et al.) and Hong-Bing Shu's group in 2012 (Zhang et al.). All of these papers

had been referenced in the introduction. We now also refer to these papers on page 8 where we describe our experiments with Ubc13.

Furthermore, we would like to highlight that the study published by Streich, 2013 which suggested that only UbcH5 isozymes are the cognate E2s for TRIM25 and TRIM32 only tested a limited number of E2s for activity (see Figure 1A of that paper) and these did not include the Ubc13/Uev1a heterodimer.

4. Page 10, first full paragraph- The data do not "indicate" but merely confirm that the Ring domain is the main catalytic unit, as this was shown by Streich et al. and subsequently others. Other conclusions in this paragraph are based on the likely spurious Ubc13 results.

We apologise, this reviewer is absolutely correct that we should have written "confirmed" here as other groups have already shown that the RING domain of TRIMs is the main catalytic unit.

This has been corrected.

With respect to the comment about the Ubc13 results, please see our reply to point 3.

5. Page 12, line 9, "Given that no dimer..."- This conclusion is unsupported by the data since it is based on negative biophysical data. The next sentence is incorrect because it is based on the irrelevant Ubc13 paralog.

As described in point 3, the experiments with Ubc13 are relevant as it is a cognate E2 of the TRIM ligases investigated here.

6. Page 15, line 5 from bottom- "by helices (on) either side..."

This has been corrected.

7. Page 17, line 8 from bottom- "...are located (on) either side..."

This has been corrected.

8. Page 18, line 6 from bottom; page 19, line 7- "MgCl₂" needs 2 as subscript, not superscript.

This has been corrected.

Referee #2:

The authors describe a focused biophysical and biochemical study of the role of oligomerization on the catalytic activity of TRIM ubiquitin ligases. TRIMs contain dimerization domains in the form of coiled-coils and are therefore often assumed require dimerisation for full activity. The authors report solution-based analyses of associations of various TRIM constructs, catalytic activity of the constructs as measured by both discharge from an E2 conjugating enzyme and in the formation of ubiquitin chains linked across lysine 63. They also determine the structures of one TRIM (32) - the solution structure of the RING of TRIM 32 is a monomer, but addition of 10 residues to the N-terminus and 9 residues to the C-terminus renders it dimeric, and this structure is crystallized. They also solve the structure of the TRIM25 RING domain in complex with a ubiquitin-bearing Ube2D1 species. This RING remains monmeric in solution but also forms a similar dimer in crystals.

Overall, the data are of high quality, the manuscript is well-written.

We thank this reviewer for their supportive and constructive comments.

One weakness is that there is not much discussion of the biology of the TRIMs - a short sentence in the introduction regarding CARDS (to the non-specialist, this is a little difficult to follow).

We have added some text to the Introduction to better explain the biology of TRIMs, especially with respect to CARDS.

I have some minor concerns that the authors could address prior to publication. In some places the writing is a bit casual - 'it is believed' comes up a lot and I'm not sure I understand the meaning. For example the statement that self-association via coiled-coil is believed to be crucial for activity - what evidence is this belief based on? Has it been reported to be crucial, or put forward as a hypothesis? Similarly B-boxes are believed to act as PPI motifs.

This is a fair point. We had used these, admittedly rather vague, statements as the literature is not always clear cut. However, we have now clarified these statements and back them up by references.

In response to the specific points raised above: it has been suggested, based on kinetic data, that self-association of TRIM32 and TRIM25 is required for catalytic activity. However, this study did not directly investigate the oligomeric state of the proteins under investigation, so the link is not direct and the requirement for oligomerisation has been implied based on Hill coefficients determined and fusion to GST. This paper has now been referenced at the appropriate point in the introduction (Streich, 2013).

B-boxes have been shown to mediate self-association in TRIM5 α and Mid1/TRIM18 and our statement about their role as protein-protein interaction motifs refers to these observations. While we had referenced relevant papers for this, the wording of this sentence was misleading and we have now rephrased it to be more specific (page 3).

The title of the section small angle x-ray scattering could be changed to indicate the findings in figure 2 rather than the technique used.

We have rephrased the title but have also moved the entire section describing the SAXS data to the end of the Results section as requested by reviewer 3.

In the comparisons of activity in figure 3 (and 5A) the different activity levels might be easier to see in a graphical format, some quantification and statistical analysis. For example the ratio of E2~ub to E2 can be used to define the 'activity' of the TRIM constructs. This would allow the representation not to include the TRIM construct itself as currently sit, asterisked which draws the attention to this band on the gels.

We agree that the overlap of the different TRIM constructs with assay components on the SDS gels of our discharge assays make it at times difficult to easily follow loss of E2~Ub and generation of E2 and Ub. This overlap of protein bands made it also impossible to directly integrate the bands from Coomassie-stained gels.

To circumvent this problem we have carried out additional discharge assays in which we used a fluorescently labelled ubiquitin (labelled with Atto 647N), thereby allowing us to integrate the fluorescent bands without interference from the E3 under investigation. Unfortunately, in this set-up it was not possible to use a ratio of E2~Ub to E2 or Ub released, which we agree would have been much better, as we see a low level of autoubiquitination of the E3 (see figure EV1) and hence the total Ub released from the E2~Ub intermediate is spread over multiple

bands. For this reason we decided to plot the loss of the E2~Ub thioester intermediate over time.

We have added graphs showing the quantification of these experiments to figures 2, 4 and 5, but have decided to also keep the “original” gels as we feel that some readers might be more familiar with this way of representing discharge assays. Similarly, we have quantified the poly-ubiquitination assays with UBE2N/Ube2V1 by supplementing the assays with 1 μ M of fluorescent ubiquitin and following the loss of mono-ubiquitin due to incorporation into chains. These assays are shown in figure 2 and EV1.

We are grateful that the reviewers asked us to quantify the discharge assays as the quantification highlighted differences in activity between constructs that were not apparent with the naked eye when analysing the Coomassie stained gels.

Also, it is a matter of preference but there are efforts in the ubiquitin fields to standardize the names of the E2s, so including Ube2D3 where UbcH5c is used, and Ube2D1 where UbcH5A is used, Ube2N in place of Ubc13, UBE2V1 would be helpful since many researchers use this nomenclature.

This is a good point and we have standardized the names all through the manuscript.

Also - is there a rationale for using one Ube2D family member for assays and a different one for crystallization? The section in methods detailing production needs careful proofing to ensure each enzyme is referred to correctly.

There was no specific rationale for using different UBE2D isoforms for assays and crystallisation and we have never observed any obvious differences between the isoforms. However, as we had to repeat the discharge assays anyway with the fluorescently-labelled ubiquitin, we have now done the assays with the same isoform that was crystallised (UBE2D1). We have also made sure that we specifically state which isoform was used in each experiment shown.

In figure 4B I found the color differences between TRIM32 and TRIM 25 difficult to follow because blue/cyan is used in figure 4A and 4C to denote monomers within one dimer.

We agree and we have changed the color of TRIM25 to yellow in the overlay. This Figure is now 3B as the SAXS section has been moved to Figure 6.

Fig 5C is referred to in the text before 5A and 5B are. Perhaps the order of the panels in figure 5 needs rethinking.

This has been changed. In addition Figure 5 is now Figure 4 as Figure 2 has been moved.

The sequence limits for each construct would be useful in the schematic in figure 1.

This has been added.

p3. that often contains protein interaction motifs should read protein-protein interaction motifs.

Some TRIMs can recognise RNA or DNA using their C-terminal domain. Hence protein interaction domain is meant in a generic sense in this sentence. Later, when we refer to B-boxes we explicitly say protein-protein interaction.

When describing RBRs as catalytic there should be reference to Wenzel (2011) Nature .

Absolutely, this has been added.

Guinier is mis-spelled as gunier in the text and in table 1.

Thanks, this has been changed.

Referee #3:

This manuscript investigates the architecture and activity of two TRIM E3 ligases - TRIM25 and TRIM32. The authors utilize a number of different constructs from each protein in a range of biochemical and structural studies. The key conclusions/observations in the manuscript are: (i) that irrespective of the oligomeric state of the RING domain in solution it appears that RING dimerization is required for ubiquitin transfer; (ii) that RING dimerisation may be stabilized by binding the E2~ub conjugate; (iii) The RING dimer traps the E2~Ub conjugate in a closed conformation that is comparable to other RINGs, however, electrostatic interactions - that resemble those between Cbl-b and the conjugated ubiquitin, are important; (iv) In the absence of E2~Ub conjugate the oligomeric arrangement of TRIMs varies, as does the impact of additional domains - i.e. in the absence of the coiled coil domain TRIM25 is a monomer, but TRIM32 is a dimer; (v) A model of how the anti-parallel coiled-coil domain of the TRIMs allows RING dimerization in the presence and absence of a higher order structure - as represented by TRIM32 and TRIM 25 respectively, is proposed.

TRIM proteins are a large family of E3 ligases that are reported to influence key signaling events in cells and this work will be of considerable interest to the wider field. The data is mostly of high quality and the conclusions well supported, but the manuscript is not easy to read and the authors should consider revising it to make the material more accessible to a wider audience. For example, it may be easier for the reader if the crystal structure of the complex is presented first, followed by the mutagenesis and then the solution studies, which together lead to a model of assembly and activity. As a minimal requirement the SAXS and NMR sections should be revised and may be better presented together.

We are pleased to see that this reviewer is supportive of our data and conclusions drawn.

We have considered restructuring the manuscript as suggested but believe that starting with the description of the crystal structure would disrupt the story as our overall aim of this study was to link the oligomeric state of a given TRIM to its catalytic activity and mechanism. However, we agree that it is easier for the reader if the SAXS and NMR sections were presented together, ideally after the structure description. Therefore we have moved these sections and accompanying figures (now combined in Figure 6) to the end of the Results section, which makes it also easier to connect our SAXS-based characterisation of the RBCC regions of TRIM25 and TRIM32 to the overall models suggested for their assembly and activity.

Major points:

1. As indicated above the SAXS data might be more meaningfully discussed after the structure solution. As presented it is a very dense read for a non-specialist, general audience as the significance of the data is not necessarily apparent at this stage

As described above the SAXS data are now presented and discussed at the end of the Results section. The NMR and SAX sections have also been reworded to make them more accessible for the non-specialist reader.

2. The assays for RING activity used throughout the paper measure either hydrolysis of the thioester bond linking the ubiquitin to the E2, UbcH5c, or formation of polyubiquitin chains by Ubc13. None of the assays are quantified - as the effect of domain deletion/mutation is sometimes modest some assays should be quantified. For Ubc13, activity is indicated as being comparable in Figure 3b but there is some indication of increased processivity for T25-RBCC - and given subsequent conclusions about dimerization/activity, this may warrant closer inspection.

We agree and as described in our reply to reviewer 2 we have now included a quantification in which we follow the loss of E2~Ub using a fluorescently labelled ubiquitin in the discharge assays or loss of mono-ubiquitin in the poly-ubiquitin chain formation assays. For details of these experiments please see our reply to reviewer 2.

Unfortunately, these assays do not allow us to make a clear statement about processivity. However, this is a very good point that warrants further investigation in future studies.

3. In figure 5 it would be interesting to analyse the V72R mutant in the linked dimer. Also the structure presented in panel 5C might be better shown first.

We agree and now show the structure first (now figure 4).

We have also analysed the V72R mutant in the linked dimer (figure 4B-D). This is an interesting mutant that shows decreased activity but not complete loss of activity.

Furthermore, we thought it would be interesting to additionally analyse the equivalent mutation (I85R) in the intrinsically dimeric TRIM32 RING domain. These experiments are now shown in figures 5C and E.

4. The authors suggest that addition of the E2~Ub conjugate triggers RING dimerization of TRIM25 RING. Can this be observed using MALS analysis?

We have carried out this experiment but unfortunately the affinity is too low to be detected by size exclusion chromatography. We can only see formation of a small proportion of the (monomeric) RING/E2~Ub complex. In contrast, when we repeat the experiment with the covalently linked RING dimer we see formation of a stoichiometric complex. These data indicate that dimerization and complex formation are intimately linked but are too weak for the isolated RING domain to be followed by SEC.

We feel that these data do not add anything to the present manuscript and we prefer to test in future studies if suitable techniques can be identified that are better suited to the quantification of this weak interaction.

5. Further, in figure 5 enhanced discharge was observed with the fused RINGs but chain building was not altered - does this reflect differences in the ability of the different conjugates to stabilize the RING dimer? If this is not the reason what other differences might account for the data.

This is indeed an interesting difference between the two E2s and we speculate that a possible reason for these differences might be the ability of Ubc13 to preferentially adopt a closed conformation even in the absence of a RING domain. We have added a sentence referring to this on page 10.

6. In general, consider reorganizing the order data appears in the manuscript so that the model presented in the discussion is better developed based on the data.

Many thanks for this suggestion. We have reorganised the manuscript as suggested, and believe that the description of the NMR data and SAXS experiments after the structure helps the flow and makes it easier to understand the model for full-length TRIM25 and TRIM32 described in the Discussion.

Minor points:

1) For TRIM25 use either RBCC or RBBCC, but not both for the long construct.

We now use RBCC throughout

2) The use of TRIM25 and TRIM32 in many figures/panels has the potential to confuse the reader. Wherever possible the figures should indicate the protein studied - ie. The MALS data in Figure 1b could be labeled with the relevant TRIM.

That's a good point. We have now added labels wherever possible.

3) Figure 4b - TRIM 32 or TRIM37? Also include relevant pdb identifiers in the text.

This is TRIM32 as labelled in the figure. We have now added the pdb identifiers to the figure legend.

4) It would be helpful to include the sequence of the crystallised TRIM domains in the extended data.

We have now added the domain boundaries of all constructs used in this study to Figure 1A as requested by reviewer 2. The sequences of the crystallised proteins are included in the pdb files that will become available upon publication and can be downloaded from <http://www.rcsb.org/>

2nd Editorial Decision

30 March 2016

Thank you for submitting your revised manuscript to The EMBO Journal.

As you can see below, the referees appreciate the introduced changes. Referee #1 has a few remaining comments that can all be addressed with appropriate text changes. Referee #3 suggests a different title - consider it. I am not so sure that I like that the title begins with unexpected, but I will leave it up to you.

Once we get the revised version in then we will accept it for publication here.

REFeree REPORTS:

Referee #1:

The Trim ligases represent the largest superfamily of Ring ligases and are characterized by a highly conserved domain architecture. This is an improved manuscript in which most of the concerns of the reviewers have been addressed, including additional data. A few points of concern remain that were not adequately addressed from the initial review:

1. page 4, paragraph 1- Streich et al. did not "propose" oligomerization was required for catalytic activity but demonstrated this point through quantitative initial rate kinetic measurements of wild type versus truncated Trim ligases. In addition, their observation of cooperative kinetics requires a priori that the active form of these enzymes is an oligomer. This point should be corrected here and elsewhere (page 17 first line) in the text.
2. page 7, paragraph 1- The reason proffered by the authors for not measuring ubiquitin conjugation is questionable. The argument that conjugation cannot be compared among the constructs of "different length...contain varying numbers of lysine residues" is not supported by experimental evidence and assumes the process is non-specific. In contrast, Streich et al. have shown that the Trims assemble both unanchored (free) and anchored (conjugates) polyubiquitin chains, the ratio of which is concentration dependant, suggesting the latter happens in trans. Moreover, the latter work demonstrates that chain formation is a valid functional readout for these enzymes. If the authors prefer to measure discharge kinetic, with its shortcomings, then state such rather than create misconceptions for the readers.
3. page 17 top paragraph- The previous study (Streich et al.) demonstrates the requirement for oligomerization (point 1 above). The authors are no doubt aware that the Hill coefficient provides a measure of the minimum number of interacting sites. Thus, a conformationally-tightly coupled tetramer (a classic concerted mechanism) yields an $n=4$ while a less tightly conformationally-coupled tetramer (closer to a sequential model) yields $n<4$, depending on the degree of coupling. Because the kinetic data is based on kinetic measurements for which the E2 concentrations (the basis for the precision in these studies) is determined empirically by stoichiometric thioester end-point assays, the $K_{1/2}$ and n values should be independent of "experimental conditions." Moreover, it is more than coincidence that the n values are consistently ca. 2-fold greater than the number of

sites seen from the structural studies. Given the crystal structure is only a single "frame" in the video that constitutes the catalytic cycle of the enzyme while the kinetics measures the rate of transit through the video, the former does not reflect the overall mechanism. The earlier kinetic work shows substrate inhibition by the E2~ubiquitin thioester, indicating sequential binding to a second functionally-distinct binding site, and mutation analysis shows binding to a non-canonical site in the C-terminal helix of the E2 that is not accounted for in the current structures. One does not need to "see" the intermediate to infer it. The combined observations indicate that the authors are consciously dismissing important aspects of the mechanism.

Referee #2:

All my concerns have been addressed.

Referee #3:

The manuscript is much improved and I am now happy for it to be published.

I wonder if the authors might consider a different title as it would be good to highlight the variable oligomerization. The following statement is taken from the abstract and some version of this might work as a title.

'Unexpected diversity in the self association mechanism of TRIMs'

Minor point:

The authors should include cc1/2 in their crystallization table.

2nd Revision - authors' response

01 April 2016

Point by point response, EMBOJ-2015-93741

“Functional role of TRIM E3 ligase oligomerization and regulation of catalytic activity“

Referee #1:

The Trim ligases represent the largest superfamily of Ring ligases and are characterized by a highly conserved domain architecture. This is an improved manuscript in which most of the concerns of the reviewers have been addressed, including additional data. A few points of concern remain that were not adequately addressed from the initial review:

1. page 4, paragraph 1- Streich et al. did not "propose" oligomerization was required for catalytic activity but demonstrated this point through quantitative initial rate kinetic measurements of wild type versus truncated Trim ligases. In addition, their observation of cooperative kinetics requires a priori that the active form of these enzymes is an oligomer. This point should be corrected here and elsewhere (page 17 first line) in the text.

We think that “proposed” is an adequate statement here as the authors of the Streich paper did not assess the oligomeric state of the proteins under investigation directly but inferred that oligomerization was required based on their kinetic analysis. (please also see our reply to point 3)

2. page 7, paragraph 1- The reason proffered by the authors for not measuring ubiquitin conjugation is questionable. The argument that conjugation cannot be compared among the constructs of "different length...contain varying numbers of lysine residues" is not supported by experimental evidence and assumes the process is non-specific. In contrast, Streich et al. have shown that the Trims assemble both unanchored (free) and anchored (conjugates) polyubiquitin chains, the ratio of which is concentration dependant, suggesting the latter happens in trans.

Moreover, the latter work demonstrates that chain formation is a valid functional readout for these enzymes. If the authors prefer to measure discharge kinetic, with its shortcomings, then state such rather than create misconceptions for the readers.

We disagree with this reviewer that our reason for carrying out discharge assays with UbcH5 isoforms is questionable. When using autoubiquitination as a proxy readout for catalytic activity there is always a problem when constructs of different length and composition are compared as the availability of lysine residues for modification is different and hence the observed activity. Discharge assays are widely used in the study of E3 ligases and are a generally accepted readout for the ability of an E3 ligase to activate an E2~Ub conjugate.

However, we agree with this reviewer that it is important to have different functional readouts and therefore we also present data analysing the formation of K63-linked polyubiquitin chains with the Ubc13/Uev1a heterodimer, which has been shown to work with both, TRIM25 and TRIM32.

3. page 17 top paragraph- The previous study (Streich et al.) demonstrates the requirement for oligomerization (point 1 above). The authors are no doubt aware that the Hill coefficient provides a measure of the minimum number of interacting sites. Thus, a conformationally-tightly coupled tetramer (a classic concerted mechanism) yields an $n=4$ while a less tightly conformationally-coupled tetramer (closer to a sequential model) yields $n<4$, depending on the degree of coupling. Because the kinetic data is based on kinetic measurements for which the E2 concentrations (the basis for the precision in these studies) is determined empirically by stoichiometric thioester end-point assays, the $K_{1/2}$ and n values should be independent of "experimental conditions." Moreover, it is more than coincidence that the n values are consistently ca. 2-fold greater than the number of sites seen from the structural studies. Given the crystal structure is only a single "frame" in the video that constitutes the catalytic cycle of the enzyme while the kinetics measures the rate of transit through the video, the former does not reflect the overall mechanism. The earlier kinetic work shows substrate inhibition by the E2~ubiquitin thioester, indicating sequential binding to a second functionally-distinct binding site, and mutation analysis shows binding to a non-canonical site in the C-terminal helix of the E2 that is not accounted for in the current structures. One does not need to "see" the intermediate to infer it. The combined observations indicate that the authors are consciously dismissing important aspects of the mechanism.

In our opinion there are currently not sufficient unambiguous experimental data available to conclusively support a direct mechanistic link between the Hill coefficients determined in the Streich study and the oligomeric states of TRIM5 α , TRIM25 and TRIM32 determined in this study and others in the literature. TRIM5 α has been shown by many groups to form higher order oligomers unless specific point mutations are introduced, which is not compatible with a Hill coefficient of 2.9 as determined in the Streich study, especially considering that this is lower than those determined for dimeric TRIM25 and tetrameric TRIM32.

The observation of a second functionally distinct binding site for UbcH5 on TRIM32 is very intriguing. However, so far we have been unable to detect such a site in direct binding studies and hence prefer not to comment on this at this stage.

The Discussion of the Streich paper concludes with the statement "Overall, the current data support a model for TRIM32 and other members of this ligase superfamily in which oligomerization is essential for catalytic competence in formation of the resulting polyubiquitin degradation signal."

In acknowledgement of this statement we have changed our text on page 17 to

"A recent kinetic study supports a model in which TRIM32, TRIM25 and TRIM5 α catalyse poly-ubiquitin chain formation through a cooperative allosteric mechanism, which would explain the requirement of oligomerization for catalytic activity (Streich et al, 2013)."

Referee #3:

The manuscript is much improved and I am now happy for it to be published.

I wonder if the authors might consider a different title as it would be good to highlight the variable oligomerization. The following statement is taken from the abstract and some version of this might work as a title.

'Unexpected diversity in the self association mechanism of TRIMs'

We considered different alternative titles but came to the conclusion that we prefer to keep the current title as it links oligomerization to catalytic activity, which we think is an important message of the paper.

Minor point:

The authors should include $cc1/2$ in their crystallization table.

This has been added.

Corresponding Author Name: Katrin Rittinger
Journal Submitted to: EMBO Journal
Manuscript Number: EMBOJ-2015-93741